# Macaque dorsal premotor cortex exhibits decision-related activity only when specific stimulus–response associations are known

Megan Wang[1,12], Christéva Montanède[2,12], Chandramouli Chandrasekaran[3,4,5,6], Diogo Peixoto[7,8], Krishna V. Shenoy[3,4,7,9,10,11,13] & John F. Kalaska[2,13]

How deliberation on sensory cues and action selection interact in decision-related brain areas is still not well understood. Here, monkeys reached to one of two targets, whose colors alternated randomly between trials, by discriminating the dominant color of a checkerboard cue composed of different numbers of squares of the two target colors in different trials. In a Targets First task the colored targets appeared first, followed by the checkerboard; in a Checkerboard First task, this order was reversed. After both cues appeared in both tasks, responses of dorsal premotor cortex (PMd) units covaried with action choices, strength of evidence for action choices, and RTs— hallmarks of decision-related activity. However, very few units were modulated by checkerboard color composition or the color of the chosen target, even during the checkerboard deliberation epoch of the Checkerboard First task. These findings implicate PMd in the action-selection but not the perceptual components of the decision-making process in these tasks.

[1] Neurosciences Graduate Program, Stanford University, Stanford, CA 94305, USA. [2] Département de Neurosciences, Pavillon Paul-G.-Desmarais, Faculté de Médecine, Université de Montréal, succursale Centre-ville, Montréal, QC H3C 3J7, Canada. [3] Department of Electrical Engineering, Stanford University, Stanford, CA 94305, USA. [4] Howard Hughes Medical Institute, Stanford University, Stanford, CA 94305, USA. [5] Department of Anatomy and Neurobiology, Boston University, Boston, MA 02118, USA. [6] Department of Psychological and Brain Sciences, Boston University, Boston, MA 02215, USA. [7] Department of Neurobiology, Stanford University, Stanford, CA 94305, USA. [8] Champalimaud Neuroscience Programme, 1400-038 Lisbon, Portugal. [9] Department of Bioengineering, Stanford University, Stanford, CA 94305, USA. [10] Bio-X Program, Stanford University, Stanford, CA 94305, USA. [11] Stanford Neurosciences Institute, Stanford University, Stanford, CA 94305, USA. [12] These authors contributed equally: Megan Wang, Christéva Montanède. [13] These authors jointly supervised this work: Krishna V. Shenoy, John F. Kalaska. Correspondence and requests for materials should be addressed to J.F.K. (email: john.francis. kalaska@umontreal.ca)

A fundamental role of the brain is to guide the physical interactions of the individual with her environment. This requires continual decisions about action choices by deliberating upon sensory evidence from the world to decide in favor of one choice over other alternative actions[1–6]. In some situations, deliberation can co-occur with decisions about action choices, for instance while trying on different pairs of boots at a shoe store before selecting one pair to buy. In others, deliberation and selection of the actions that implement the decision can be dissociated in time, for instance by choosing which pair of boots to buy on the store's web site before going to the store to purchase them. Of course, we can also make a categorical decision about whether or not we like a particular pair of boots without any intention to buy them.

Many premotor brain areas are implicated in sensorimotor decision-making processes[1,7–12]. The oculomotor system has been extensively studied using tasks in which subjects chose between saccade targets in known locations by estimating the net direction of visual motion in random-dot kinematogram (RDK) stimuli[13]. Putative neural correlates of the deliberation process leading to a saccade were found in multiple cortical and sub-cortical saccade-related structures[10,12,14–18]. This indicates a broadly distributed intentional framework for saccade decision-making in which neural correlates of the decision process arise in premotor areas that guide actions to report the decision[2,4,9,19,20]. An ongoing challenge is to determine whether and to what degree the temporally evolving activity contributes to the process of deliberation on sensory evidence or with the choice of action. With rare exceptions[21–23], that determination is uncertain because the subjects knew how a perceptual choice would be mapped onto action choices before the salient sensory input began. As a result, the perceptual-deliberation and action-selection processes could occur simultaneously.

Similar issues arise for arm movements. The dorsal premotor cortex (PMd) is a region in which sensory instructional and action-related information converge to guide voluntary arm movements[2,24–41]. PMd neural activity can covary with the physical properties of sensory inputs that inform motor responses[42,43] and generate representations of potential reach choices[44]. PMd activity can also covary with higher-level abstract action-related concepts before action choices are fully specified, such as the general goal of a future action[45] or a visuomotor task rule[46], and can even express learned stimulus–response associations during mental rehearsal without overt movement[47].

Our labs have been studying the role of PMd in reach decisions in tasks in which subjects must discriminate the dominant color of a multi-colored checkerboard decision cue to select between two color-coded targets. In our initial studies, we used a Targets First (TF) task in which the stimulus–response associations, indicated by the color of the targets, were known in each trial before the checkerboard appeared[26,48,49]. We both found that neural activity in PMd of monkeys was correlated with checkerboard color "coherence" (see Methods) and direction of the chosen target[26,49,50]. Nevertheless, the TF task shares the same interpretational limitation of many prior studies in that checkerboard onset can initiate the perceptual-deliberation and action-selection processes simultaneously.

To address this task confound, we both implemented a task variant, the Checkerboard First (CF) or Checkerboard First with Delay (CFD) task, in which the sensory decision cue is presented before the stimulus–response associations are revealed. This permitted human subjects to make a categorical perceptual decision about the dominant checkerboard color independent of how the decision would be reported[48]. We sought PMd neural correlates of decision-making processes, defined operationally here as differential neural activity that predicted any of the monkeys' task-related decisions, including the dominant color of the checkerboard and the direction or the color of the chosen target.

A key hypothesis was that if PMd reflects perceptual decision-making independent of action choices, then neural activity during the Checkerboard-observation epoch of the CF and CFD tasks would reflect the checkerboard's dominant color, its critical property that determines the monkeys' action decisions. We found that virtually no PMd unit showed a differential response to the checkerboard's dominant color or to the color of the chosen target. This indicates that PMd does not express correlates of the critical color perceptual decision process that informs the reach target choices in the TF or CF/CFD tasks. Instead, PMd units become differentially active only when complete information about the stimulus–response associations that determine action choices is available, and their activity primarily reflects the properties of those actions (e.g., reach direction), the strength of evidence supporting those actions, and the temporal dynamics of the action decisions.

## Results

**A task designed to dissociate perceptual and action decisions in time.** Two male macaque monkeys (T and Z) performed variants of a sensorimotor decision-making task (Fig. 1a). They had to determine the dominant color of a multi-colored checkerboard and report that color by reaching to the corresponding colored target. In a given session, target locations were fixed and presented in opposite directions from the center hold. As a result, both monkeys could potentially anticipate the spatial locations of the targets at the start of each trial. However, target colors were assigned randomly on each trial, so the monkeys could not know which target would be which color until they appeared.

The TF task followed the event timeline used in many sensorimotor decision tasks, in which the response options (targets) appeared before the decision cue (checkerboard)[17,18,26,48,49]. As a result, deliberation about dominant checkerboard color could occur concomitantly with planning for the reach, because each color was already associated with a specific target location. Crucially, in the CF and CFD tasks, the order of the two visual cues was reversed. The checkerboard appeared first, and the monkeys could deliberate upon its dominant color[48] but could not prepare a specific reach to report it until the colored targets appeared. The monkeys were free to choose when to initiate a reach after the second cue appeared in each task. Details of the task event sequence and timing varied between the two laboratories (Fig. 1a; see also Methods).

Figure 1b, c (left) show T's task performance during all neural recording sessions. To facilitate comparison of task performance, Z was tested in separate behavioral sessions without neural recordings, using seven checkerboard color coherences ranging from 4% to 80% (Fig. 1b, c, right). A reduced set of coherences (4%, 20%, and 100%) was used during neural recordings with Z; task performance was completely consistent with the trends described here (Table 1).

The performance of both monkeys was strikingly similar in the TF task, even though Z's checkerboards contained a third, task-irrelevant color and half of its viewed checkerboards were dynamic (see Methods and Supplementary Discussion). Both performed at above-chance levels with 4% checkerboards (T: 63.0%; Z: 71.6%) and plateaued at ~100% success rates with checkerboard coherences of 20–90% (Fig. 1b, blue curves). The psychometric curves were symmetric and centered on 0% color coherence, indicating that neither monkey showed a major color bias. Reaction times (RTs) were longest for the 4% checkerboards (T: 585 ± 6 ms (s.e.m.), $n = 41$ sessions; Z: 583 ± 36 ms, $n = 33$

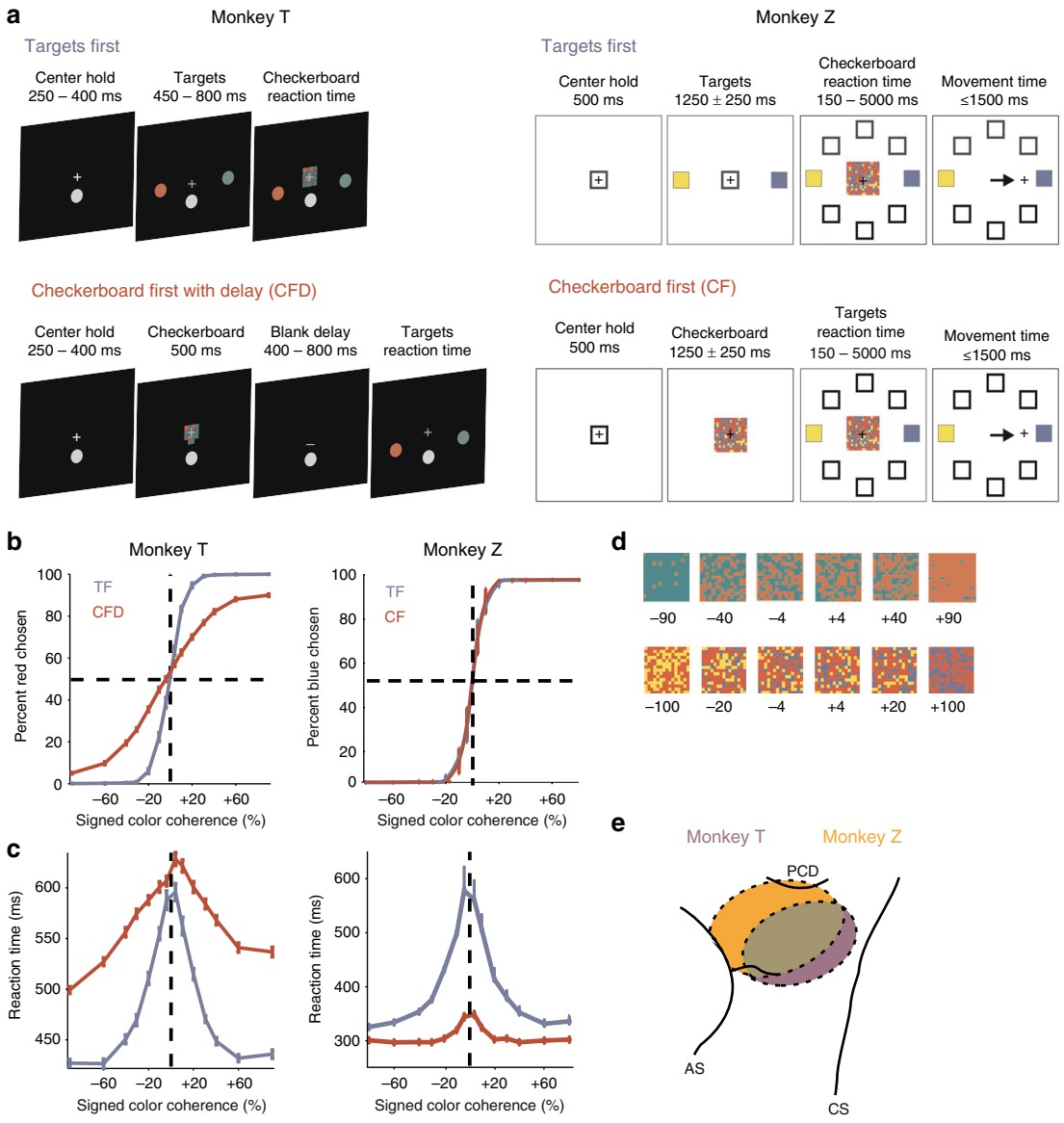

**Fig. 1** Choice behavior of two monkeys (T and Z) in the Targets First(TF) and Checkerboard First (CF) tasks. **a** Two monkeys from two labs performed variants of a checkerboard color discrimination task. The TF task design (top row) follows the usual convention of presenting possible action choices—here, colored targets—before a decision cue—here, a colored checkerboard. The goal is to report the dominant color of the checkerboard by reaching to the corresponding colored target. Note that in this design, perceptual decision-making (checkerboard color discrimination) and action selection (choosing and reaching to the target with the matching color) both occur in the reaction-time interval between checkerboard onset and movement onset, and in theory can happen simultaneously. In the CF and CF with Delay (CFD) task design (bottom row), the order of presentation is reversed to dissociate perceptual decision-making and action selection. Target color identities are assigned randomly on each trial. **b** Psychometric curves of the probability of red (T) or blue (Z) target color choices (mean ± s.e.m.) as a function of signed checkerboard color coherence (see (**d**) for examples). As the strength of color evidence in the checkerboards supporting a reach to a red/blue target increased, the probability of reaches to those colored targets increased. **c** Chronometric curves of reaction times (RTs; mean ± s.e.m.) as a function of signed checkerboard color coherence. As the strength of color evidence in the checkerboard increased, RT durations tended to decrease. See also Supplementary Figure 1. **d** Examples of red-green checkerboards for T (top row) and blue-yellow checkerboards for Z (bottom row), labeled with their level of signed color coherence (units are %). **e** Recordings were performed in dorsal premotor cortex (PMd) in the hemisphere contralateral to the reaching arm. Data include units recorded from single electrodes and linear arrays. This schematic map illustrates the approximate locations of recording sites based on stereotactic coordinates. Histology has not yet been done on either monkey. PCD precentral dimple, AS arcuate sulcus, CS central sulcus

sessions) and became systematically shorter as coherence increased, approaching an asymptote at minimal values at 60–90% checkerboards (T: 435 ± 3 ms for 90% coherence, Z: 333 ± 9 ms for 80% coherence; Fig. 1c, blue curves).

The performance of the two animals differed in the CF/CFD task. T did not achieve 100% performance even at the strongest

color coherence (lapse rate 7.4% for 90% coherence), and the psychometric curve was shallower in the CFD task (Fig. 1b left, red curve) than in the TF task. In contrast, the psychometric curves for Z were largely similar in the two tasks (Fig. 1b, right, red curve), possibly in part because of the longer Checkerboard-observation period (1000–1500 ms) in the CF task and its

**Table 1 Mean success rate (%) and mean RT ± s.e.m. (ms) for the data files collected during all 104 neural recording sessions in Z**

| Checkerboard coherence (%) | 4 | 20 | 100 |
|---|---|---|---|
| *TF task* | | | |
| Success rate (%) | 66.1 | 95.9 | 100.0 |
| Mean RT ± s.e.m. (ms) | 529.1 ± 18.2 | 426.3 ± 11.6 | 323.7 ± 4.5 |
| *CF task* | | | |
| Success rate (%) | 67.2 | 97.3 | 100.0 |
| Mean RT ± s.e.m. (ms) | 341.5 ± 8.1 | 303.5 ± 6.3 | 298.2 ± 5.6 |

continued presence after the targets appeared until the monkey initiated a reach.

To compare psychophysical thresholds for checkerboard color coherence between the two tasks, we fit the folded psychometric curves (% correct as a function of unsigned coherence) to cumulative Weibull functions[10]. For T, the psychophysical threshold was 10.4% in the TF task, and increased to 41.9% in the CFD task. For Z, psychophysical thresholds were lower and very similar between tasks (TF: 7.0%; CF: 6.8%).

The monkeys also showed differences in their RT trends in the CF/CFD tasks. T's RTs were systematically longer in the CFD task than in the TF task at all checkerboard coherences (Fig. 1c, left, red line). This was most pronounced for the checkerboards with the strongest color coherence, even though T only performed those trials at 92% success rates. In contrast, Z was faster in the CF task than the TF task at all coherences, with the largest reduction for checkerboards with the lowest coherences (Fig. 1c, right, red line). There was a much smaller dependence of RTs on checkerboard coherence in the CF task compared to the TF task; RTs for 4% checkerboards (350 ± 13 ms) were only ~45 ms longer than for the 80% checkerboards (303 ± 7 ms).

Z was trained and tested in the CFD task (see Methods) and also showed a decrease in success rates for checkerboards with stronger color coherences, with a lapse rate of 10.5% (Supplementary Figure 1A). However, unlike T, Z continued to show nearly all the temporal savings in the CFD task that had been observed in the CF task. Its RTs for the 4% checkerboards (348 ± 8 ms) were essentially identical to that in the CF task, and were only slightly prolonged for the 80% checkerboards (328 ± 7 ms) compared to the CF task (Supplementary Figure 1B).

These behavioral findings indicated that both monkeys made use of the sensory information available during the Checkerboard-observation period of the CF/CFD tasks. Although T's RTs were prolonged rather than reduced in the CFD task, its choice behavior showed that it retained some unknown but task-salient information about the checkerboard during the memory-delay period and used that stored information to make target choices whose success rates increased and RTs decreased systematically with the evidence strength of the no-longer-visible checkerboard. Z showed very similar psychophysical curves in the TF and CF tasks, but also had an overall marked reduction in RTs in the CF and CFD tasks that was largest for checkerboards with the weakest color coherence, as in human subjects[48]. Shorter RTs in the CF/CFD tasks might be a behavioral sign that Z made a categorical perceptual decision about the dominant color of the checkerboards during the Checkerboard-observation period.

We next asked how and when neural correlates of the perceptual and action decisions were expressed in PMd as a function of the amount of information available about specific action choices. Strong correlates with the color of the chosen target or with the amount of color evidence in the checkerboard independent of action choices would indicate that PMd expresses activity reflecting the process of perceptual deliberation on the task-salient sensory evidence. Neural correlates with the direction of the chosen target or with the amount of color-independent evidence favoring an action choice would support a role in acquiring evidence supporting specific action decisions.

**Recordings in PMd and unit selection criteria.** We recorded neural data from PMd contralateral to the performing arm, including the left hemisphere of T using single microelectrodes and single linear multi-contact electrodes, and both hemispheres in Z using single microelectrodes (Fig. 1e). For both animals, units recorded using single microelectrodes were well-isolated putative single neurons. During linear-array recordings in T, neural activity was routinely recorded simultaneously from multiple electrode contacts; the large majority of that activity was from putative single neurons while the remaining units were multi-unit clusters (see Methods). We recorded from 499 units in T during the CFD task, of which 351 units were also recorded during the TF task. For T, the two target locations were always to the left and right of the starting hand position and thus not in the optimal task-related preferred and non-preferred directions for most units. Indeed, this is not possible for the linear-array recordings during which many units were typically recorded in each session. T's units were selected for analysis if they responded during any epoch of the task. In contrast, all recorded units in Z were pre-screened for task-related responses using eight-direction 1-Target and 2-Target instructed-delay tasks[44,49]. Complete data sets were then collected from 104 isolated units in both the TF and CF tasks (41 and 63 units from the left and right hemispheres, respectively). Only one unit was recorded per session, and the targets were placed in each cell's task-related preferred movement direction (PD) and diametrically opposite direction as assessed in the eight-direction tasks. Thus, T's units were expected to have less directional selectivity overall in the task since the target locations were fixed for all sessions. In contrast, Z's units should have stronger directional selectivity because target locations were adapted to each units' task-related directional tuning.

**Units exhibited heterogeneous responses to first visual cues.** We briefly describe how PMd units responded during observation of the first visual cue in each task, when only partial (in the TF task, color-coded targets) or no (in the CFD/CF task, colored checkerboard) specific action-choice information is available.

The example unit from T in Fig. 2a, c showed no change in activity during the first-cue observation period in either task, which continued into the memory-delay period of the CFD task. This unit was representative of the large majority of units in T. The example unit from Z (Fig. 2b, d) also showed no evident response during the Targets-observation period in the TF task, but exhibited a small fluctuation in activity 100–300 ms after checkerboard appearance during the Checkerboard-observation period of the CF task. We performed a bin-wise search to identify significant rapid changes in activity in response to the first cue (see Methods). Only 1/351 and 0/499 units in T showed this change in response to the first cue in the TF and CFD tasks, respectively. In contrast, many units in Z responded shortly after the appearance of the first visual cue in the TF (42/104) and CF tasks (52/104); 35/42 and 44/52 of those responses were detected in the time period 100–300 ms after the cues appeared (Figs. 2 and 3, Supplementary Figure 2). These responses in Z may be transient representations of potential actions[44], and may be more prominent in Z because the targets were always placed in each unit's preferred and non-preferred directions (see Supplementary Discussion).

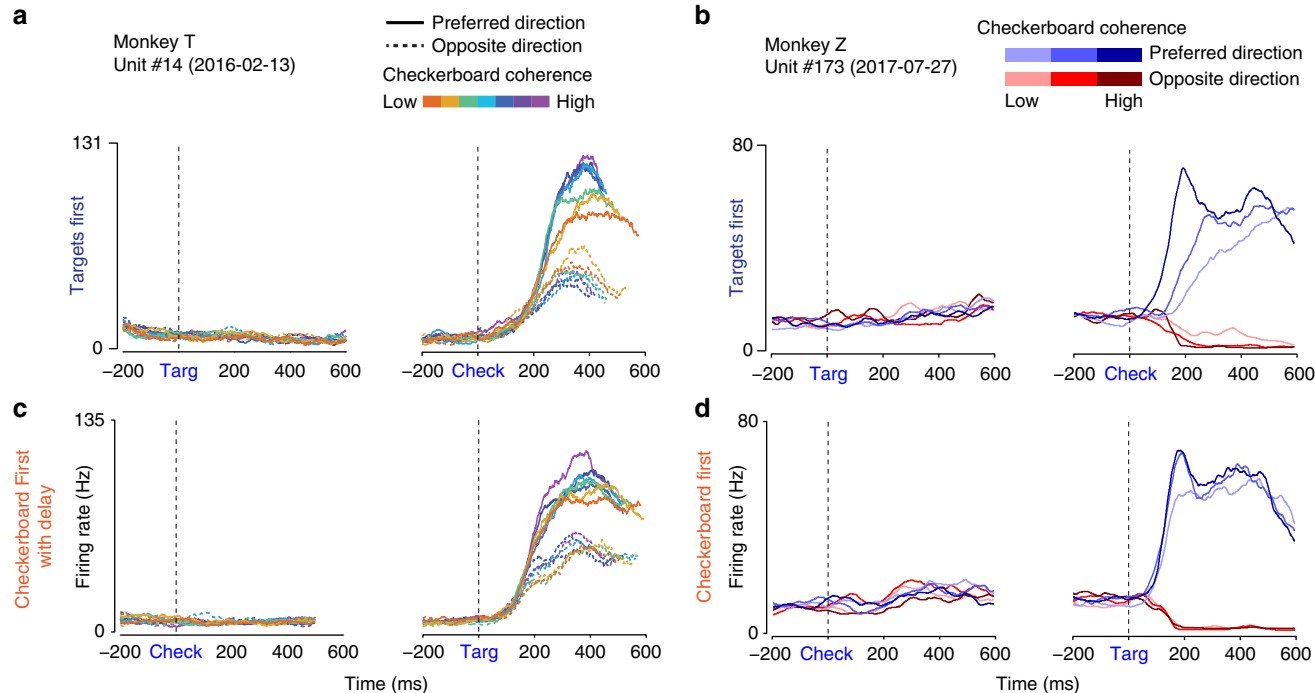

**Fig. 2** Example single-unit activity profiles in the TF and CF/CFD tasks. Data are aligned to the appearance of the first visual cue (left) and the second visual cue (right) in each task. **a**, **b** TF task. Consistent with previous work with similar tasks, the rate of change in neural activity after the appearance of the checkerboard cue (Check) in the TF task correlates with both checkerboard coherence and reach direction. Neither unit responded to the appearance of the first visual cue, the Targets (Targ), in that task. **c**, **d** CFD/CF task. In the CF and CFD tasks, the same units show a reduced effect of checkerboard color coherence on the rate of change of activity after the Targets appear. The unit from T showed no change in activity during the initial Checkerboard-observation period of the CF task, but the unit from Z showed a small rapid response change, that was detected 280 ms after Checkerboard appearance (see text)

**Units had differential decision-related responses after second visual cues.** The second visual cue in each task gave the monkeys the missing sensory information needed to complete the sensorimotor decision process and select a reach target. Task-related units typically expressed their strongest responses at that time. Unit activity in the TF task in both monkeys (Figs. 2 and 3) was similar to our previous findings, including differential activity dependent on the direction of reach and on the coherence level of the checkerboards[26,49]. In the CFD task, the example unit from T had a smaller range of rates of change of discharge in both reach directions as a function of the coherence of the no-longer visible checkerboard. The example units from Z showed nearly identical rapid rates of change in activity for the 100% and 20% checkerboards and only a modestly slower rate of change for the 4% checkerboards (Figs. 2 and 3; Supplementary Figure 2).

To quantify these responses, we estimated the slope of the choice selectivity signal associated with each checkerboard coherence during a time window 0–300 ms after the appearance of the two visual cues in the two tasks[26,51] (see Methods). The slopes of the choice selectivity signals after the first cue appeared in each task were very low in both tasks (Fig. 4a, c). This indicated that the activity during this epoch did not predict the action direction choices of the monkeys, even while observing the color-coded targets in the TF task. No change in firing rate as a function of checkerboard coherence was seen in the TF task, as should be expected because the checkerboard had not yet appeared. In contrast, color coherence information was visible during the initial Checkerboard-observation epoch in the CF/CFD tasks but no information about how to link a color to a reach, and the mean choice selectivity slope also did not vary with coherence.

The slopes of choice selectivity signals after the second visual cue appeared were large and varied systematically with checkerboard coherence, so that activity differentiating the chosen reach directions increased more quickly with higher coherences (Fig. 4b, d). This was more prominent in the TF task than the CF/CFD task. The increase in single-unit slope values in the TF task was statistically significant between the 4% and 20% coherences (two-tailed paired $t$-test; T: $p = 8.35E{-}11$, Z: $p = 1.05E{-}11$), and between the 20% and highest-coherence checkerboards (two-tailed paired $t$-test; T: $p = 1.23E{-}06$, Z: $p = 6.59E{-}09$). Slope values were higher in Z than in T, in part likely reflecting the sampling bias in Z to maximize the directionality of recorded unit activity.

For T, the slopes of single-unit choice selectivity signals units in the CFD task increased at higher coherence (slopes at 20% vs. 90%, two-tailed paired $t$-test; $p = 3.08E{-}04$) but not at lower coherences (slopes at 4% vs. 20%, two-tailed paired $t$-test; $p = 0.7302$). Furthermore, the slopes were generally lower in the CFD task than in the TF task, especially at higher coherences (two-sample $t$-test comparing slopes across tasks; for 4%, $p = 0.05$; for 20%, $p = 1.4E{-}03$; for 90%, $p = 7.83E{-}04$). In contrast, the choice selectivity signal slopes for Z in the CF task increased significantly from the 4% to the 20% checkerboards (two-tailed paired $t$-test; $p = 1.45E{-}05$), but were similar for the 20% and 100% checkerboards (two-tailed paired $t$-test; $p = 0.26$). Moreover, the slopes were significantly higher in the CF task than the TF task for the 4% checkerboards, non-significantly higher for the 20% checkerboards, and significantly lower for the 100% checkerboards (two-sample $t$-test comparing slopes across tasks; for 4%, $p = 1.05E{-}05$; for 20%, $p = 0.011$; for 90%, $p = 5.6E{-}04$).

In summary, PMd activity after the second visual cue reflected the direction of chosen reach in both tasks. The rate of change of

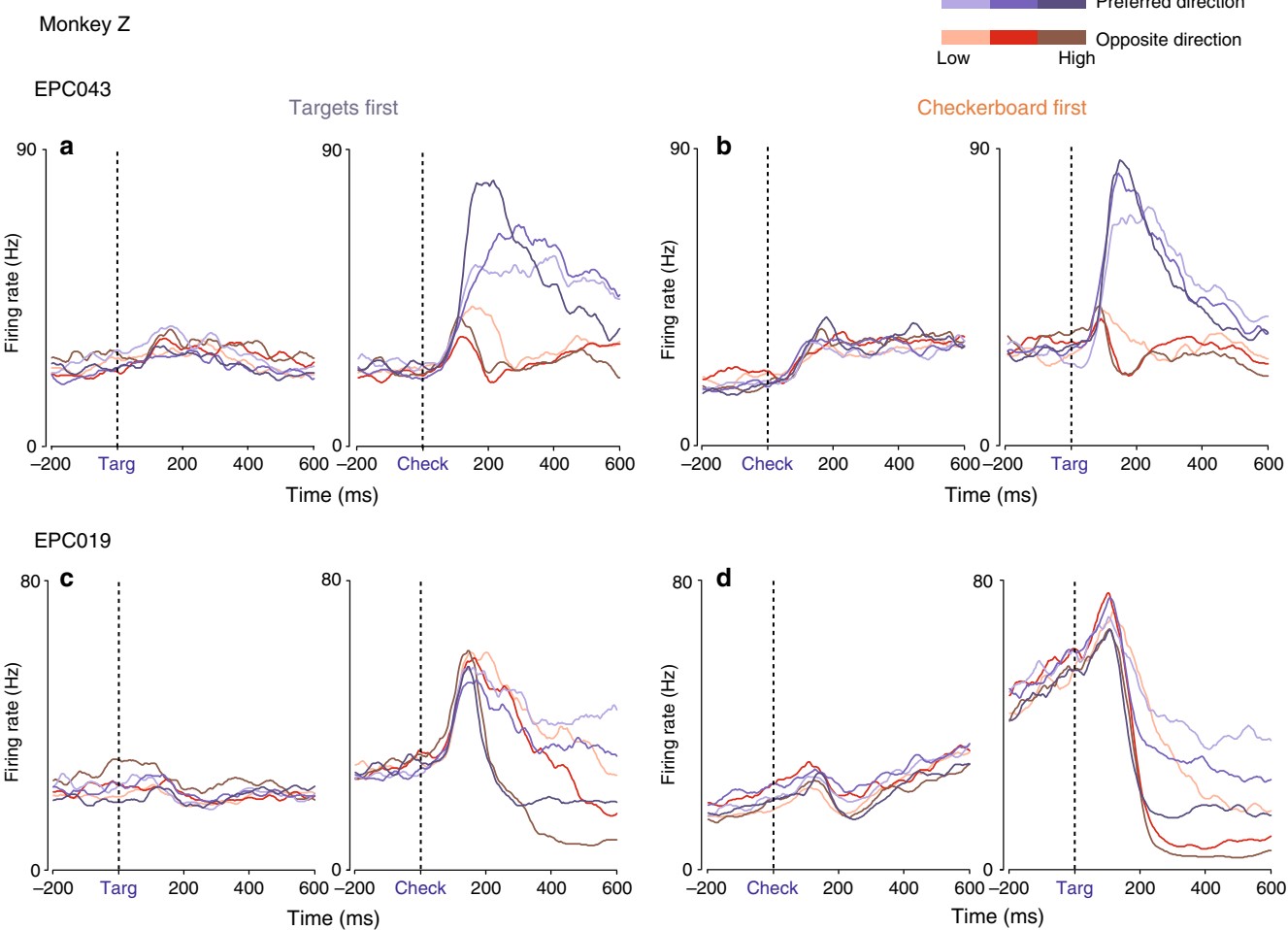

**Fig. 3** Example units with responses to the first visual cues of each task. See also Supplementary Figure 2. **a**, **b** Unit EPC043 emitted a transient increase in activity detected at 160 ms after the Targets appeared in the TF task (**a**), and a stronger sustained response beginning 140 ms after the appearance of the checkerboard in the CF task (**b**). **c**, **d** Unit EPC019 emitted a small rise in activity that was detected at 240 ms after Targets appearance in the TF task (**c**). In the CF task (**d**), there was a brief increase in activity detected at 140 ms after the appearance of the checkerboard, followed by a transient suppression and then a pronounced ramp increase for the remaining duration of the Checkerboard-observation period

the choice selectivity signal was strongly dependent on checkerboard coherence in the TF task in both monkeys, but was reduced in the CF/CFD tasks.

**PMd neural responses are expressed in action-decision space.** We next used linear regression to understand how time-varying neural activity covaried with different task variables. We aligned single-trial data (in sequential non-overlapping 20 ms time bins) to the onset of each visual cue in both tasks. For each unit, we regressed the binned single-trial firing rates on the following predictor variables: chosen reach direction, chosen target color, signed color coherence (the amount of evidence for one color over the other), and signed directional coherence (the amount of color-independent evidence supporting a reach direction). The last predictor requires knowledge of the stimulus–response mapping of that trial. We included both correct and incorrect choices so that the color and direction of the chosen target reflects the monkeys' interpretation of the sensory evidence rather than the correct dominant color of the checkerboard (see Methods).

Almost no unit showed a significant covariation with any of the four predictors at any time during the Targets-observation period of the TF task (Fig. 5a, b). In contrast, after checkerboard onset, the proportion of units that reflected the direction of the chosen

target (green) rose rapidly, and a somewhat smaller proportion of units reflected the signed strength of the color-independent directional evidence towards a given target (dark blue). These directional response components were more prominent in Z than in T, as expected. Critically, very few units were significantly modulated by the color of the chosen target (magenta) or the signed checkerboard color coherence (turquoise) at any time after the checkerboard appeared.

Similarly, almost no unit showed a significant correlation with the color of the chosen target or the signed color coherence of the checkerboards at any time in the trial in the CF/CFD task (Fig. 5c, d), including the initial Checkerboard-observation period. After the targets appeared, the proportion of units with effects of chosen reach direction rose rapidly for both monkeys. Significant correlations with signed directional coherence did not appear during the Checkerboard-observation period in either monkey, but emerged after the targets appeared, even though the checkerboard was no longer visible for T. The incidence of the directional coherence effects was lower than in the TF task in both monkeys. The short 20 ms bin size may have biased this analysis against detecting significant effects in units with low firing rates. However, these findings were confirmed and complemented by a repeated-measure ANOVA using firing rates

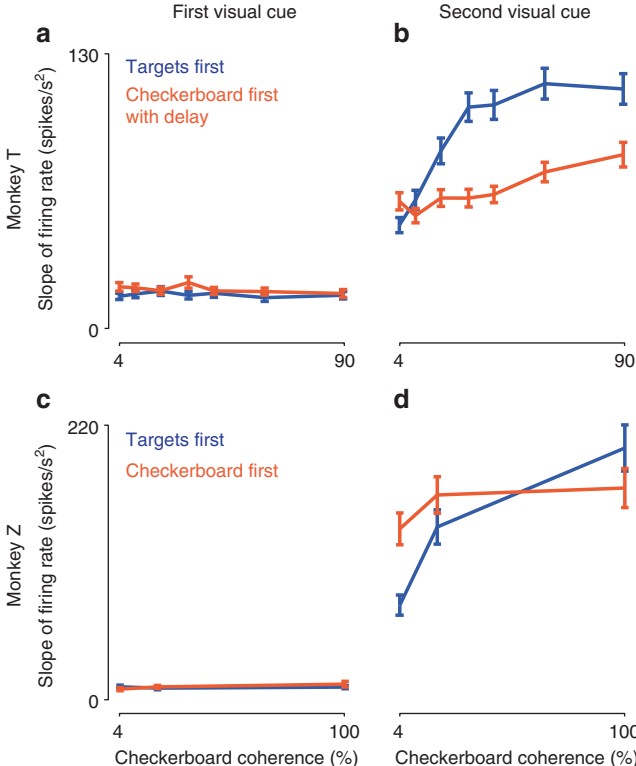

**Fig. 4** Neural activity reflected action choice and unsigned checkerboard coherence following the second visual cue. Plotted are the discharge rate slopes (mean ± s.e.m.) of the choice selectivity signal as a function of checkerboard coherence. The choice selectivity signal is the difference in activity for the two opposite reach directions. For each unit, the choice selectivity signal recorded 0–300 ms after the appearance of the first visual cue (**a**, **c**) and the second visual cue (**b**, **d**) was calculated separately for trials with each checkerboard coherence in the TF (blue) and CFD/CF tasks (orange), and then fit to a linear ramp function to estimate the short-latency rate of change of activity in response to the two visual cues (see Methods). While observing the first visual cue, choice selectivity signal slopes were small and did not vary as a function of checkerboard coherences in both tasks in T (**a**) and Z (**c**), even while observing the checkerboards in the CFD/CF task. This indicated that the first cue did not evoke a predictive directional signal in neural activity, as should be expected. Following the appearance of the second visual cue in each task however, strong choice selectivity signals appeared in PMd in both T (**b**) and Z (**d**). Slopes of the signals were strongly modulated by checkerboard coherence in the TF task (blue) but much less sensitive to checkerboard coherence in the CFD/CF task (orange) in both monkeys. This would be expected if the sensory evidence about the dominant color of the checkerboard had been processed before the colored targets appeared in the CFD/CF tasks and then stored in a form that was more amenable for a binary decision about action choices

averaged across entire epochs of interest. There were strong main effects of chosen reach direction and unsigned evidence strength after the second cue appeared in each task, but very few main effects of checkerboard dominant color (Supplementary Figure 3; Supplementary Table 1).

In summary, we emphasize three main findings. (1) The chosen direction of movement independent of all other task factors was by far the strongest predictor of PMd activity in both monkeys in both tasks, and was only expressed after the monkeys had received both visual cues and could apply the stimulus–response mapping to choose a target. (2) The next strongest predictor was a color-blind neural response component that was likewise correlated with the spatial direction in which the monkeys reached; this predictor also reflected the combination of information provided by the two visual cues but was graded by the strength of the directional evidence provided by the checkerboard after applying the stimulus–response mapping. This is the component that most strongly implicates PMd in the action-choice process based on sensory evidence. It arose only after the monkeys received both cues, and appeared at the same time as the activity explained by the chosen direction predictor in the TF task. Interestingly, it was also present in the CF/CFD task, but was weaker and arose more slowly in both monkeys, suggesting that the strength of the evidence provided by the checkerboard was at least partially discounted by the time the colored targets appeared. (3) There were almost no significant correlations with either chosen target color or signed color coherence of the checkerboards, at any time in either task. That is, there was no prominent neural correlate in PMd of the perceptual deliberation process that was essential to identify the checkerboard's dominant color and correct target color.

**Units encoded reach direction sooner in the CF/CFD task**. Another important question is whether the order in which the cues were presented had an impact on the onset latency of choice-related activity of PMd units in the two tasks. We performed an ROC analysis of the ability of an ideal observer to predict the dominant checkerboard color or the chosen target direction based on neural activity in sequential non-overlapping 20 ms time bins. Strong population-level signals about target choices in both monkeys appeared only after both visual cues appeared in each task (Fig. 6). As expected, this target-choice information was stronger in Z's units compared to T's (cf., Figs. 4 and 5), because target locations were at the task-related preferred movement axis of each unit in Z but were in two fixed spatial locations for T. In contrast, the ability of an ideal observer to discriminate the checkerboard's dominant color remained at baseline during the Checkerboard-observation epoch of the CF/CFD task in both monkeys, and remained at baseline after the second visual cue appeared in both tasks in Z. There was a very modest but statistically significant increase in the discriminability of the checkerboard dominant color in the population activity of T after targets appeared in the CFD task but not after the checkerboard appeared in the TF task (Fig. 6).

To estimate the onset of a significant increase in the AUC values of the ROC analysis, we tested the distribution of values across the population at each time point after the appearance of the second visual cue, compared to the distribution of values at baseline, 200 ms before the second cue appeared (Wilcoxon signed-rank test, one-tailed, see Methods). The latency was defined as the first time point at which the distribution of AUC values for the post-cue activity was significantly different from the baseline values for 50 consecutive milliseconds. For both animals, the latency for reach direction detection was shorter in the CF/CFD task than the TF task (T, TF: 206 ms, CFD: 183 ms; Z, TF: 193 ms, CF: 134 ms), even though T's behavioral RTs were longer in the CFD task than the TF task. To determine if this difference in directional latencies between tasks was significant, we created null distributions of latency differences using a bootstrap method (see Methods). Our observed inter-task latency differences (T: 23 ms; Z: 59 ms) were greater than any of the null distributions of 1000 latency difference values generated by bootstrap-resampling and random re-assignment of neural data to the two tasks[52] (significance based on bootstrap distribution, T: $p = 0.001$; Z: $p = 0.001$).

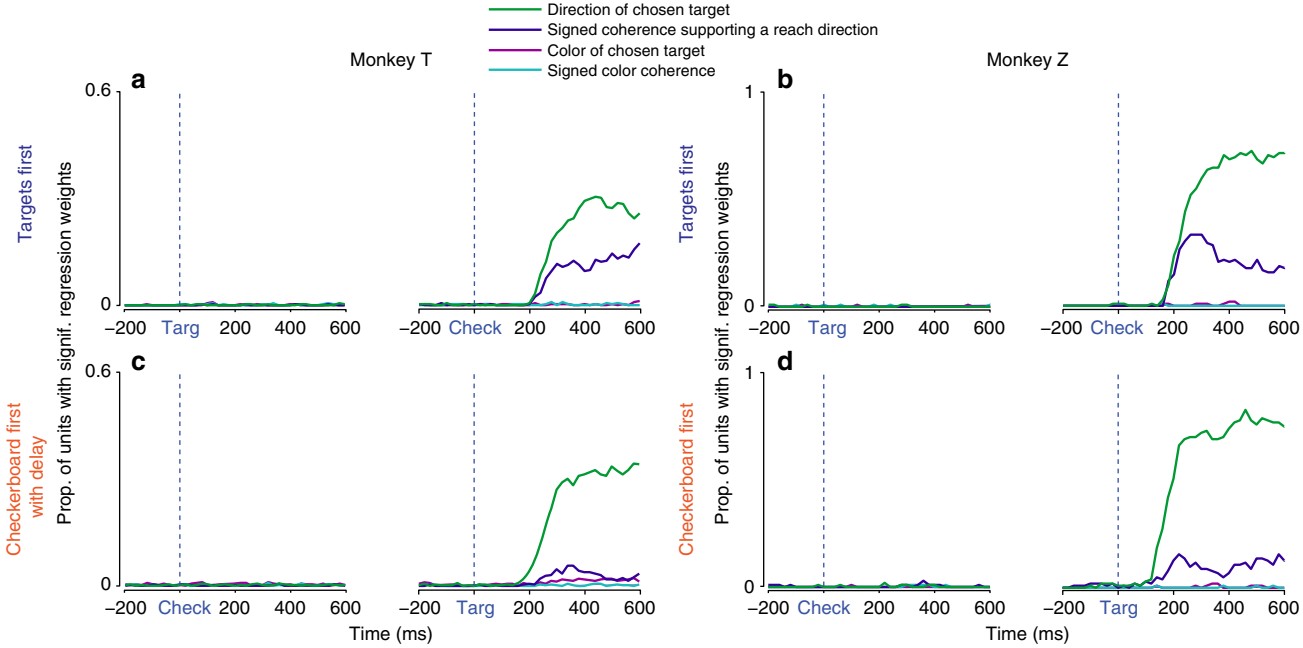

**Fig. 5** PMd activity reflected action decisions, not perceptual decisions. Each unit's firing rate at a given 20 ms time point across trials was regressed on a linear model with the following predictor variables: direction of the chosen target (green), signed checkerboard coherence favoring a particular reach direction independent of its color (dark blue), color of the chosen target independent of its direction (magenta), and signed checkerboard color coherence independent of the direction of the chosen target (turquoise). This regression analysis was repeated for all units and all time points from −200 ms before to +600 ms after the appearance of the first visual cue in each trial (left of subfigure) and of the second visual cue (right). Plotted are the proportions of units with significant regression weights for a given predictor at each 20 ms time point, for each monkey (columns) and task (rows). Very few significant correlations with any regression predictor were seen during the observation period of the first visual cue (left panel of each pair) in either TF (**a**, **b**) or CF/CFD (**c**, **d**). In particular, significant correlations with chosen target color (magenta) and signed checkerboard color coherence (turquoise) rarely occurred at any time in either task in either monkey. However, following the second visual cue, there were significant correlations with the direction of the chosen target (green) and the signed level of checkerboard evidence favoring a target direction independent of its color (dark blue). Correlates with the direction of chosen target (green) were comparably frequent across both tasks for each monkey, with more units exhibiting selectivity in Z because the targets were placed in each unit's preferred and non-preferred directions. The incidence of significant correlations with signed checkerboard evidence favoring a particular reach direction (dark blue) was present, but substantially lower, in both monkeys in the CF/CFD tasks compared to in the TF task

Shorter onset latencies in the CF/CFD task than the TF task further support the conclusion that some information about the checkerboard color composition was processed during the Checkerboard-observation period in both monkeys. This suggests that the temporal dynamics of neural responses in PMd following the appearance of the second visual cue are different in the two tasks, resulting in a reduction of the onset latency of the earliest PMd activity predicting action choices in the CF/CFD task.

## Discussion

We investigated how task demands affect the extent to which neural correlates of perceptual decision-making are present in PMd. In many decision-making studies, the stimulus-response associations are specified before the decision cue is presented[10,17,18]. Similarly, in the TF task of this study, the appearance of the two color-coded target cues at the start of each trial provided the specific stimulus-response mapping before the checkerboard appeared. In contrast, in the CF/CFD task, we inverted the task timeline by presenting the checkerboard before the color-coded targets. The checkerboard and targets each provided different partial information for the forthcoming reach movement, and the monkeys could not choose a specific reach action until they received both pieces of information.

Consistent with our previous results[26,49], PMd activity expressed differential decision-related correlates after checkerboard onset in the TF task, including the level of the color-

independent checkerboard coherence supporting a reach target (signed directional coherence) and the ultimate action choice, but not the physical color composition (signed color coherence) of the checkerboard or the color of the chosen target. Similar effects were observed in the CF/CFD task after the targets appeared. Importantly, while the monkeys observed the checkerboard cue of CF/CFD tasks, PMd did not express any differential decision-related activity that predicted their perceptual choices.

**PMd's role in perceptual decision-making**. Perceptual decision-making has been described as the process by which an individual commits to a proposition on the basis of perceived sensory information; that decision is typically reported in animal studies by a differential motor response[4,6]. Of course, as already noted, we make many decisions every day ("I like those boots") without committing to a particular action, unless "inaction" is a motor decision. In this study, a categorical perceptual decision about the dominant color of a checkerboard stimulus is combined with the information about the spatial configuration of colored targets to arrive at a motor decision. The critical novel finding of the present study is that the appearance of the checkerboards in the CF/CFD task did not elicit any color-specific correlates of either the sensory evidence (signed color coherence) or a categorical decision about the dominant color during an ongoing perceptual decision process. These results confirm that PMd is implicated in the processing of color-independent information supporting

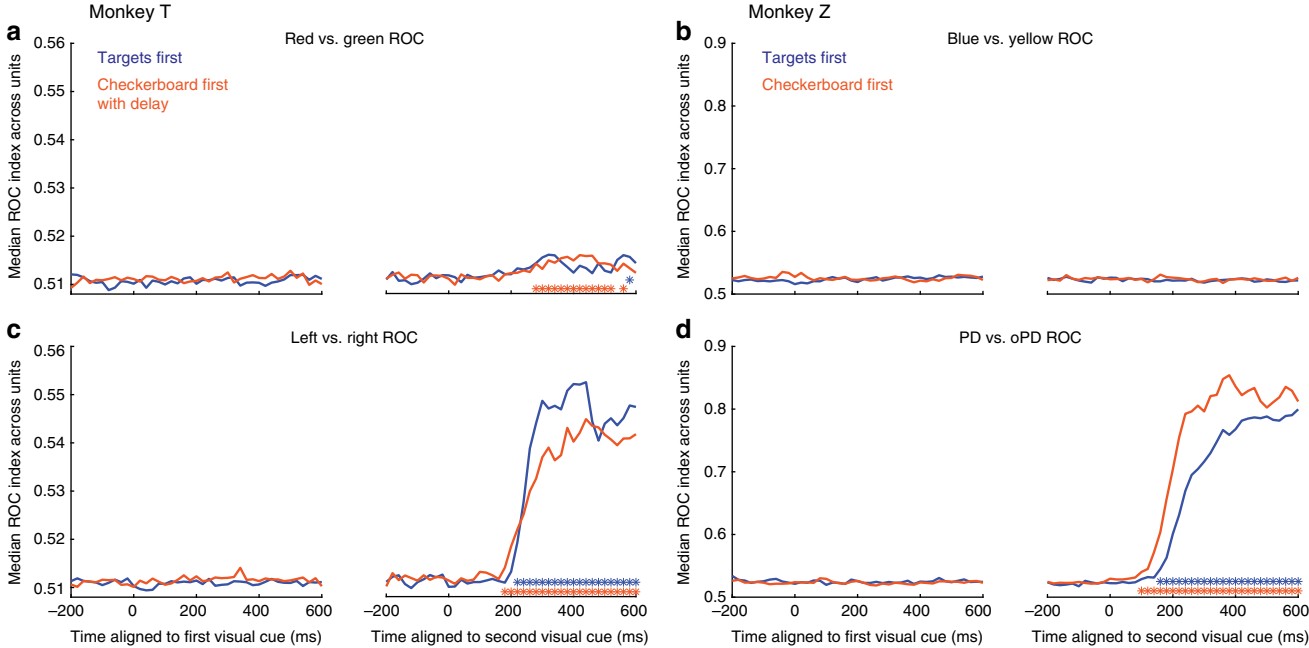

**Fig. 6** Identifying color selectivity and reach direction selectivity. Receiver operating characteristic (ROC) analysis was performed on the single-trial discharge rates for each unit at every 20 ms time point from −200 ms before to +600 ms after the appearance of the first (left of each pair of figures) and second (right) visual cue in each trial, for trials sorted according to the dominant color of the checkerboard (**a**, **b**) or the chosen direction of reach (**c**, **d**) in the TF (blue) and CFD/CF (orange) tasks. The median area-under-the-ROC-curve (AUC) value across the population of units is plotted at each 20 ms time point. The median AUC values for the Color test were very small at all times in both tasks in both monkeys (**a**, **b**), indicating a very weak representation of information about the dominant color of the checkerboards. In contrast, there was an abrupt increase in median AUC values for Direction shortly after the appearance of the second visual cue in both tasks in both monkeys (**c**, **d**). As in previous directional analyses, the direction-related ROC values were larger in Z than in T. To detect a significant increase in the detectability of color or direction information in the population activity, the distribution of AUC values measured in each 20 ms time step was compared to a baseline 20 ms interval −200 ms before the appearance of the first visual cue (one-tailed Wilcoxon signed-rank test, p < 0.01). Time steps in which the distribution of AUC values were significantly higher than the baseline distribution are indicated by blue and orange asterisks for the TF and CFD/CF tasks, respectively

action choices and reveal that PMd primarily processes the information provided by the visual instructional cues about the spatial attributes of action choices. Furthermore, it expresses correlates of the differential action-related decision process only after the monkeys have received all the information from both cues required to map checkerboard color onto a specific target choice. The near absence of significant correlates of the critical physical property of the cues—color—that informs the final action choice indicates that PMd does not make a substantial contribution to the non-motor perceptual aspects of these tasks.

These findings also constrain PMd's role in the motor decision. The PMd units may be receiving a time-varying signal about the most likely color of the checkerboard but are directly transforming that information into a signal about the most likely direction of reach, once the color-location conjunctions are known. Alternatively, the PMd units may be receiving a more abstract signal about the mounting evidence for the solution of the color/location matching rule based on the color information in the checkerboard, and are translating that into a signal about the most likely target location for a reach independent of its color. Consistent with this hypothesis, Mante et al. [53] have shown that prefrontal cortex neurons in and near the frontal eye fields (FEF) encode the color information in colored RDK stimuli and convert it into evidence for the choice of a color-coded saccade target in a task that is conceptually similar to the TF task. Finally, PMd may be receiving a time-varying signal about the most likely location of the target after all salient information has been processed and is only implicated in the preparation of the chosen action.

However, this last possibility seems to be unlikely given all the previous studies that have implicated PMd in the conversion of sensory information into abstract goals and specific actions[2,25,26,30,31,33–35,40,41,44–46,49,54,55].

**Comparison of findings in the two monkeys in the TF, CF, and CFD tasks.** The behavioral and neurophysiological results were very similar in both monkeys in the TF task. Interestingly, T's behavior in the CFD task was inconsistent with that of Z and human subjects in a CF task[48]. Its RTs were longer than in the TF task, even for the strongest checkerboards, and T made lapse errors for those stimuli. There are several possible contributing factors: (1) the 500 ms Checkerboard-observation period may have been too short to complete a categorical perceptual decision on every trial; (2) the Checkerboard-observation period may have been too short to form an accurate short-term memory of its physical features during the 400–800 ms delay period; or (3) the memory of the checkerboard or of the perceptual decision may have decayed during the delay period. The first possibility is unlikely, given that T's success rates were systematically higher in the TF task at all checkerboard coherences even in trials with RTs (and thus checkerboard viewing times) shorter than 500 ms (Supplementary Figure 4A). Regarding the second and third possibilities, T's success rates in the CFD task were modestly higher for shorter compared to longer delay periods, as well as when longer checkerboard durations were tested (Supplementary Figure 4B-D).

T's strategy may have been to store a memory trace of some features of the checkerboard and largely defer the decision process until the colored targets appeared in the CFD task. A perceptual decision might be formed in CFD-like tasks by sequential sampling from working memory[56]. Alternatively, T may have at least partly formed a perceptual decision before the targets appeared, but the dynamics of its motor decision were still sensitive to the strength of the sensory evidence that had informed the perceptual decision, for instance by being modulated by confidence in the correctness of its perceptual decision[57–60]. Of course, this is entirely speculative because we did not ask T to report a metacognitive estimate of its confidence in each decision.

In contrast, Z was substantially faster even for low coherence checkerboards in a CFD task (Supplementary Figure 1). Nevertheless, it showed some of the same costs as T for the brief fixed Checkerboard-observation period and memory delay; its RTs for the strongest checkerboard color coherences were modestly prolonged by ~25 ms compared to that in the CF task, and it had a lapse-error rate of ~10% for high-coherence stimuli. These costs in both monkeys are somewhat surprising. Easy color discriminations such as those for checkerboards with 80–100% coherence should be very rapid (30–50 ms[61,62]). Similarly, PMd neurons covary with differential action choices based on the color of an instructional cue (e.g., go/nogo; reach toward or away from a visual cue) in only ~50–100 ms[35,63,64]. The imposed delay period seems to have impeded the ability of both monkeys to retain information about the dominant color of the checkerboard or to use it correctly to identify the reach target after they appeared. This effect was most apparent for the nominally easiest checkerboards in both monkeys.

Many PMd units in Z responded to the appearance of the first cue in each task (Figs. 2, 3, Supplementary Figure 2). These responses may contribute to the overall target selection process[2,44], but they did not encode the color information in either the targets (TF task) or the checkerboard (CF task), or display any differential decision-making signals that predicted either the target color or direction choices of the monkey (Figs. 4– 6). This is consistent with prior reports of non-differential activation of premotor cortex neurons before the final action was specified[34,44,45,65,66] that may reflect the likelihood of future potential actions[67].

Despite these behavioral and neural differences, our primary findings were remarkably robust in both monkeys—PMd did not strongly encode the critical color dimension of the instructional cues and did not show neural correlates of differential decision-making processes until the monkeys had received the information provided by both instructional cues.

**Comparison with other perceptual decision-related areas.** Other motor areas such as the superior colliculus[23] (SC), pre-supplementary and cingulate motor areas[68], and especially lateral intraparietal cortex[21,69] (LIP) have shown different degrees of correlations to perceptual decisions versus motor actions, by presenting the sensory evidence before the target choices, and by decoupling the mapping between the evidence and action choices. For instance, by inducing saccades with intracortical micro-stimulation, Gold and Shadlen[22] found behavioral evidence of developing oculomotor commands in the FEF directed towards known target locations either in the direction of RDK motion (pro-saccade task) or the opposite direction (anti-saccade task) while the monkeys observed RDK stimuli, but not in a colored-target task in which red and green targets appeared at random locations only after the RDK stimuli were extinguished. The results implicated FEF primarily in the motor aspects of the sensorimotor decision process,

and only when the stimulus–response associations were known, consistent with the present findings.

Horwitz et al.[23] presented saccade targets at unpredictable locations after RDK stimuli were extinguished. Some SC saccade-related neurons responded during RDK motion; their activity reliably predicted that the monkey would ultimately choose the target that signified that they had perceived RDK motion directed towards the neuron's preferred movement field, even though the report saccade direction was not yet known. This activity was considered a potential mnemonic representation of the perceived directionality of the sensory evidence on which the monkeys would base their motor decision, expressed in the spatial framework of saccade movements in SC circuits.

Finally, Bennur and Gold[21] used a variant of the colored-target task[22] in which they revealed stimulus–response mappings before, during or after RDK stimuli were presented. The first and last are conceptually similar to our TF and CF tasks, respectively. LIP neurons expressed neural correlates of all salient sensory and motor aspects of the sensorimotor decision leading to saccade direction choices, including target colors and the perceived RDK motion direction before the metrics of the report saccades were known (see also ref. [70]). Freedman and colleagues likewise found explicit representations of sensory properties, cognitive decisions and motor reports in LIP[3,69,71]. These results suggest that the parietal cortex may be tightly involved at the intersection of sensory and motor processing by expressing features of the salient sensory evidence explicitly in its activity, whereas PMd does not in our tasks.

**Comparison to previous PMd findings.** Our findings are consistent with previous studies showing that PMd activity reflects the action-related information provided by sensory cues that guide motor goals. Wise and colleagues[25,55] showed that PMd responses to a visual instructional stimulus are strongly modulated when it signals different motor responses or a shift in attention rather than a movement. Similarly, in an instructed-delay match-to-sample task in which the color of the first cue signaled which of two buttons to press after colored stimuli appeared above them, PMd neurons did not signal the color of the first cue, whereas PFC did[31]. At a more abstract level, PMd neurons can signal whether a future reach will be to the leftward or rightward of two targets, independent of the physical identity of instructional cues, even if the subject does not know exactly where on the screen the two targets will appear[45]. One possibility is that PMd represented this abstract goal via a relative spatial encoding mechanism, similar to the spatial mnemonic strategy suggested by Horwitz et al.[23]. This may also be a more abstract form of the representation of the spatial location of potential reaching targets in a 2-Target instructed-delay task[44,49]. Neural responses in the 2-Target task also showed only very modest correlations with the colors of the instructional cues[44,49]. All of these findings indicate that PMd is predominantly implicated in processing the spatial information about future action choices provided by instructional cues, but does not strongly express neural correlates of the salient physical properties of the instructional cues.

There is also some evidence for PM involvement in signaling the stimulus–response mapping itself. Wallis and Miller[46] trained monkeys to report whether two sequentially presented images were the same or different by releasing a key with their hand either immediately after receiving the second image, or after a further 500 ms delay. A rule cue presented with the first visual image provided the stimulus–response mapping for each trial (whether same ("match") or different ("mismatch") images were reported with immediate key release). Very few PM neurons

responded differentially to the identity of the visual images, but many differentially signaled the same/different rule.

Finally, Romo and colleagues studied PMd in tasks in which monkeys reached to one of two closely spaced buttons to report the relative frequency differences of two sequentially presented tactile stimuli[5,11] or to report whether the stimuli had the same or different temporal structure[42,43]. In the relative-frequency task, few PMd neurons (~10%) showed correlates with the frequency of the two stimuli, and even fewer correlated with the chosen button-press action, suggesting that the neural population sampled in that study was not strongly implicated in either the perceptual or motor aspects of that task. In contrast, ~20–35% of a different PMd sample population in the temporal-structure study[42,43] expressed a differential categorical signal about properties of the first tactile stimulus, other neurons signaled the specific ordinal sequence of the different stimulus pairings after the second stimulus was presented, and ~20–30% signaled the categorical same/different decision independent of their actual structure. The last neurons could also be a potential correlate of the final motor decision because of the fixed stimulus–response mappings, reminiscent of the findings by Nakayama et al.[45]. It is also noteworthy that these studies[5,11,42,43] used tactile stimuli whereas the present study and others that failed to find prominent PMd correlates of the physical properties of instructional cues[25,31,45,49,55] all used visual stimuli.

**Summary**. When the perceptual assessment of a checkerboard decision cue could be made in the context of known specific stimulus–response mappings onto action choices (TF task), PMd units generated a differential decision-related signal reflecting the final action choice and the strength of the sensory evidence supporting the correct reach direction, but not the critical physical dimension of the checkerboard, its dominant color. When this link was broken and perceptual decisions about the checkerboard could be formed before the specific action to report the decision was known (CF/CFD task), PMd did not respond in a differential decision-related manner to the checkerboard itself. Explicit representations of the color/location conjunctions of the targets or of the color composition of the checkerboard, that are required to perform these tasks, were not found in PMd, and are presumably expressed elsewhere. Dorsolateral PFC is a leading candidate[31,53], and is currently under study in these tasks.

## Methods

**General Information**. Two rhesus macaque monkeys (*Macaca mulatta*), T (9-year-old male, 15.0 kg; same T as in Chandrasekaran et al.[26]) and Z (9-year-old male, 12.0 kg), were used in the study. T's home environment, standards of care, and this experiment were approved by the Stanford University Institutional Animal Care and Use Committee. Z's housing, veterinary care and experimental protocols were approved by the institutional animals-in-research committee (CDEA—Comité de déontologie de l'expérimentation sur les animaux, *Université de Montréal*), and respected all institutional and national guidelines.

**General task design**. We used two main variants of a decision-making task in which subjects chose between two opposite reach directions based on the content of two successive visual stimuli that provided different types and amounts of sensory evidence supporting each reach choice in each trial. In both variants, the goal for the subject was to determine the dominant salient color of a checkerboard-like visual stimulus, and to report that color by making an arm reach to the corresponding colored target. The difficulty of the sensorimotor decision was manipulated by varying the relative numbers of squares of two task-relevant colors. This was roughly equivalent to varying the coherence of dot motion in RDK tasks. However, RDK tasks require detection of a coherent-motion signal against a random-motion noise background. In contrast, in the checkerboard stimuli, each evidence element (a colored square) is easily detected and discriminated. The challenge is to assess their relative numbers to estimate the dominant color of the checkerboard. We use the term color coherence here to indicate the degree to which the task-salient squares in the checkerboard are the same color or not. Signed color coherence is the difference in number of colored squares for each category (e.g. # red squares–# green squares) divided by the total number of task-

relevant squares in the checkerboard. Thus, if all the task-salient squares are of the same color, the color composition of the checkerboard stimulus is said to be 100% coherent whereas a checkerboard with equal numbers of the colored squares has 0% color coherence. Positive values signify predominantly Red (T) or Blue (Z) checkerboards, and negative values signify predominantly Green (T) or Yellow (Z) checkerboards.

A key differentiator for this study compared to RDK tasks is that the color of an object does not have any inherent association with any parameter of a reach movement, such as target spatial location or reach direction. Color only becomes action-relevant in our tasks by application of an arbitrary stimulus–response mapping rule; the subjects decide on the dominant color of the checkerboard and then use an operantly conditioned color-location matching rule to associate it with the target of the same color. This is not the case in RDK stimuli, which have an intrinsic physical property—the spatial direction of coherent dot motion—that is also usually directly mapped onto the direction of the motor output. A second differentiator is that typical RDK stimuli are stochastic and dynamic, with variable numbers of short life-time dots moving in the coherent and random directions from frame to frame. In contrast, all checkerboard stimuli observed by T and half the stimuli presented to Z were static, with each square remaining visible and stationary for the entire duration of the Checkerboard-observation period. Moreover, the illusion of motion evoked by the RDK stimuli is experienced by observation of a sequence of static dot images presented rapidly across time. In contrast, the color of each square in a checkerboard image should be discriminable after a single brief presentation[62]. A fourth differentiator is that low-coherence RDK stimuli usually contain a small number of dots that move coherently in only one of the two opposite directions and the primary perceptual challenge is detecting that weak unidirectional signal in the random-motion noise. In contrast, low-coherence checkerboards contained large but nearly equal numbers of easily discriminable colored squares that each unequivocally supported one or the other of the two action choices.

The TF task variant followed the event timeline used in many sensorimotor decision tasks[17,18,26,48,49]. First, two color-coded targets appeared, providing the subject with sensory information about the two reach choices[44] and how color would be associated with reach direction. The checkerboard appeared later. Deliberation about dominant checkerboard color could occur concomitantly with planning for the reach, because each color was already associated with a specific target location. The monkeys were free to initiate a reach to a target at the time of their choosing after checkerboard appearance. Note that each monkey only ever had two colors to choose from (red and green targets for T, blue and yellow targets for Z). In addition, Z had to ignore a third color (red) that was present in all of its viewed checkerboards and that by coincidence was one of the two task-salient colors for T (further details below).

Crucially, in the Checkerboard First (CF) and CFD tasks, the order of the two sensory events was reversed. The checkerboard appeared first, but the monkeys did not yet know which color would be associated with a given target location and reach direction. Thus, the monkeys could in theory deliberate upon the checkerboard's dominant color, but could not prepare a specific motor response to report it.

Details of task structure and recordings varied between the two laboratories, as detailed below.

The following five sections describe methods used with T at Stanford University.

**Experimental setup**. Throughout the experiment, T sat in a primate chair (Crist Instruments, Snyder Chair) ~30 cm in front of an LCD computer monitor (Acer HN274H, then Acer XG270HU) on which the task would be presented. The animal's non-reaching (left) arm was loosely restrained with a tube and cloth sling. The stimulus presentation and data collection were controlled by a custom computer system (MathWorks' xPC Target and Psychophysics Toolbox). We placed a photodetector (ThorLabs PD360A) in the corner of the computer screen to detect the onset of various task events to a 1 ms resolution. Hand position was measured by taping a reflective bead (11 mm, Northern Digital Inc.'s Passive Spheres) to the tip of the middle finger of the reaching (right) hand, and tracking the location of this bead in three-dimensional space using an infrared tracking system (Polaris Spectra; Northern Digital Inc.). Eye position was tracked using an infrared camera (ISCAN ETL-200 Primate Eye Tracking Laboratory) mounted overhead; the eye image was reflected to the camera above using an infrared mirror (ThorLabs) placed at a 45° angle in front of the animal's nose. The infrared mirror allows visible light to pass through, so it does not obstruct the animal's view of the computer monitor.

**Task design**. In the Targets-First (TF) task (Fig. 1a), T initiated a trial by placing its right hand on a center hold circle (24 mm diameter) and fixating its gaze on a cross (6 mm diameter), located above the center hold circle. Once these two conditions were met, there was a brief variable delay of 250–400 ms, and then two monochromatic targets (one red, the other green) were presented 100 mm to the left and right of the center hold. These targets were presented for 450–800 ms, during which the animal maintained center hold and eye fixation (Targets-observation period). Finally, a static checkerboard stimulus containing variable numbers of red and green squares from trial to trial was presented, centered at the

fixation cross, and served as the go cue for T to make its report (Checkerboard-RT epoch). It was free to initiate its chosen reach action as soon as it was ready. The moment that center hold or eye fixation was broken at the onset of its reach response, the checkerboard disappeared but the targets remained visible.

In the Checkerboard-First with Delay (CFD) task (Fig. 1a), the presentation order of the targets and checkerboard were reversed. The trial began in the same way as in the TF task, and the center hold delay was the same at 250–400 ms. Then, the checkerboard appeared for a fixed period of 500 ms (Checkerboard-observation period), and subsequently disappeared for a memorized-delay period of 400–800 ms, after which the colored targets appeared. The appearance of the targets served as the go cue for T to make its report (Targets-RT epoch). As in the TF task, the monkey could initiate its reach movement as soon as it made its target choice.

The size and locations of all visual stimuli were the same in both tasks, and the red and green colors for the targets and checkerboard were identical and isoluminant (22 cd m$^{-2}$, Konika Minolta). The assignment of red and green to left and right targets was randomized between trials. The checkerboard consisted of a $15 \times 15$ grid of 2.5 mm × 2.5 mm squares. The task difficulty was adjusted by varying the number of red and green squares in the checkerboard (Fig. 1d). For each dominant color (red or green), we used seven difficulty levels (# non-dominant squares and # dominant squares: $11 + 214$, $45 + 180$, $67 + 158$, $78 + 147$, $90 + 135$, $101 + 124$, and $108 + 117$). These levels correspond to coherence levels (difference in red and green squares, divided by the total number of squares) of 90.2%, 60%, 40.4%, 30.7%, 20%, 10.2%, and 4%. In each trial, a single static checkerboard matrix was presented in which the R and G squares were distributed randomly within the 225-square checkerboard matrix. A different random matrix was presented on each trial, even within the same checkerboard coherence. All task factors (dominant checkerboard color, checkerboard coherence, and correct colored target location) were presented in a randomized sequence. Data were collected until neural isolation was lost (single-electrode recordings) or until the monkey was sated. Typical daily data sets comprised roughly 2000 trials of CFD task only, or 1000 trials each of CFD and TF tasks.

**Training history**. T was first trained to make reaching movements towards targets on the computer screen, for pieces of fruit and then for juice reward. It was then trained on the TF task, starting with the highest checkerboard coherence (90.2%). At the beginning, a high-coherence checkerboard was presented before the targets were presented; once the association between checkerboard color and target color was learned, the order was inverted to present the TF. Further details can be found in ref. [26]. To train T on the CFD task, we began by presenting only the highest checkerboard coherence with a 300 ms delay between checkerboard presentation and target presentation. We gradually increased the delay and added gradually lower checkerboard coherences, over many daily training sessions. T did not experience targets at any location other than left or right of the central start position.

**Recording chamber implantation and neural data collection**. T had an acrylic head implant with a recording chamber over left PMd/M1 (coordinates A16, L15; Fig. 1e). In this recording chamber (19 mm diameter), a series of small burr holes (3 mm diameter) were drilled sequentially as needed across the entire recording period through the acrylic implant and skull to access dura and brain. The neural data were recorded using either single electrodes (22 sessions) or linear arrays (19 sessions). Single electrodes were FHC tungsten electrodes #UEWLGCSEEN1E (Frederick Haer & Co, Bowdoin, ME, USA). Linear arrays were Plexon (Dallas, TX, USA) U probes with platinum–iridium recording sites, 16 channels spaced 150 μm apart (specifically: PLX-UP-16-15ED-150-SE-100-25(640)-15T-700). Single electrodes were lowered into the brain until a unit was found; linear arrays were lowered until all electrode sites were in brain, preferably with a unit on the deepest and shallowest electrodes. Single neural records were recorded during single-electrode sessions but multiple neural records were routinely collected simultaneously during linear-array sessions. The units were sorted online using BlackRock Central software. Units were included for detailed analysis if they were responsive at any time during the trial. Data were collected in blocks of roughly 500 trials per task and alternated between blocks of TF and CFD tasks in sessions in which both tasks were used.

**Neural data pre-processing**. To identify putative single units, we examined the inter-spike interval (ISI) distributions for each unit[72]. Spike timing information was collected at 30,000 samples/s. We considered ISI violations to be those ISI that were <2 ms, a conservative refractory period between action potentials. A unit was considered a single unit if it had <1.5% ISI violations. Of the 499 units collected in CFD task, 441 (88.4%) units were identified as single units, with a mean of 0.44% ISI violations; the remaining multi-unit records had a mean of 3.48% ISI violations. Of the 351 units collected in TF task (all of which overlap with the CFD units), 310 (88.3%) units were identified as single units, with a mean of 0.46% ISI violations; the remaining multi-unit records had a mean of 2.72% ISI violations. Of these 351 units, 304 were consistently classified as single units across both tasks and 33 were consistently classified as multi-unit records across both tasks. The remaining 14 cells were not consistently classified across both tasks, which could occur due to gradual drift during the recording session.

Firing rates were constructed by convolving a 50 ms acausal box car filter with spike times at 1 ms resolution. The exception is for Fig. 2a, c, in which we used a 75 ms box car filter for better visualization. Behavioral reaction time in each trial was calculated as the time at which hand velocity exceeded 10% of the maximum hand velocity. Data shown include correct and incorrect trials (depending on the analysis), with reaction times >300 ms, and do not include overt change-of-mind trials in which the monkey began to reach to one target and then reversed direction and completed a reach to the opposite direction[36,73].

The following five sections describe methods used with Z at University of Montreal.

**Experimental setup**. Where T used its arm to touch targets on a monitor screen, Z used a pendulum-like handle that moved over a horizontal digitizing tablet (hand position measurements at 100 Hz, ±0.05 mm precision; for technical details of the task apparatus, see ref. [48]), to displace a 6 mm cross-shaped cursor between targets displayed on a vertical computer monitor at a viewing distance of 60 cm.

**Task design**. The TF task structure for Z was identical to the Choose-and-Go task used in previous studies[48,49]. Each trial began when a small open white square (1.0 cm) appeared at the center of the monitor (Fig. 1a). The monkey used its arm and pendulum to position the on-screen cursor in the central square and held it there for 500 ms. Two monochromatic square target cues (4.5 cm; one yellow and one blue) then appeared at opposite sides of the central square (15.5 cm separation between the centers of the target squares) for 1250 ± 250 ms (Targets-observation period). The same two opposite target locations (out of eight possible locations arranged in a circle) were used for each block of trials for a given unit according to its reach-related directional tuning, but varied from unit to unit (see below). After a variable period of 1250 ± 250 ms, the central square was replaced by the checkerboard stimulus, and white squares appeared at the other six target locations in the task, serving as the go signal (Checkerboard-RT epoch). Z was free to reach to the chosen target at any time, without an imposed pre-reach delay. The checkerboard stimulus disappeared as soon as the cursor position exited the boundary of the original small central target. Z had to reach the target within 750 ms and stay within the target for 1000 ms to receive a liquid reward if the chosen target was correct.

The checkerboard consisted of a $15 \times 15$ grid (4.0 cm) that contained a total of 100 yellow and blue squares plus 125 task-irrelevant red background squares (Fig. 1d). For each dominant color (B or Y), three difficulty levels were used ($0 + 100$, $40 + 60$, and $48 + 52$) during neural recordings, corresponding to 100%, 20%, and 4% levels of checkerboard coherence. In half of the trials, checkerboards were static, while in the other half they were dynamic—a new checkerboard matrix with the same numbers of colored squares but different square positions was displayed every 50 ms. Static versus dynamic stimuli had little or no systematic impact on the psychophysical performance of human subjects[48] or on the task performance and neural activity recorded in two other monkeys[49]. All task factors (correct target location, correct target color, checkerboard color coherence, and static/dynamic checkerboards) were presented in a fully balanced randomized-block sequence. A complete task file comprised 120 correctly performed trials (2 targets × 2 colors × 3 coherence levels × 2 checkerboard conditions × 5 replications). If the monkey chose the incorrect target in a given trial, that trial was re-inserted into the remaining pseudo-random trial sequence until all combinations of trial conditions were completed successfully, resulting in data files containing 120 correct trials and variable numbers of incorrect trials.

Z performed a version of the CF task without a memorized-delay period (CF task, Fig. 1a). Its temporal structure was identical to the TF task here, except that the checkerboard cue was presented first during a Checkerboard-observation period after the initial center start-target period. At the end of that observation period, the two color-coded target cues appeared on opposite sides of the checkerboard cue, along with white target squares at the six other target locations (Targets-RT epoch). The monkey could initiate its reach choice as soon as it had made its decision. The checkerboard disappeared as soon as the cursor exited the boundary of the original small central target. Data file structure was identical to the TF task. No gaze fixation control was imposed at any time in any of the tasks for Z[44,49,54].

Z also performed a memorized instructed-delay task with a single target cue presented at the beginning of the delay period (1-Target task, 1T[44,49]), and a memorized instructed-delay task in which two color-coded potential target cues were presented simultaneously in opposite directions in each trial, followed by a monochromatic central color cue that unambiguously signaled the correct target in each trial (2-Target task, 2T[44,49]). Z also performed a version of the TF task that included an extra imposed pre-reach delay period. In this Targets-first with Checkerboard-Delay (TFCD) task, each trial began like the TF task. However, at the end of the initial Targets-observation period, the Checkerboard cue appeared for 1750 ± 300 ms, while the target cues remained visible. The monkey was not allowed to make a reach during this Checkerboard-observation period. At the end of that pre-reach delay period, white squares appeared at the other six target positions as a go signal and the monkey could make its chosen reach movement. Data file structure was identical to the TF task. Neural data collected from these three tasks will not be presented here.

**Training history**. Z was first acclimatized to sit in a custom-made primate chair, and then trained in a standard eight-direction center-out reaching task without delay periods, for juice rewards. It was then trained to perform the 1T task, followed by the 2T task. Following this training, we sequentially introduced the TF, CF, and TFCD task variants using multi-colored checkerboard stimuli (Fig. 1d). In each task variant, Z first performed the tasks with only the 100% checkerboards, followed by the 20%, and then the 4% checkerboards as performance improved and stabilized. After first learning the tasks with the right arm, neural data were collected from the left PMd/M1. Z was then trained to perform the tasks with the left arm and neural data were collected from the right PMd/M1.

To facilitate comparison of task performance of the two monkeys, Z was also tested in the TF and CF tasks with seven checkerboard coherence levels (4%, 10%, 20%, 30%, 40%, 60%, and 80%) in daily sessions separate from neural recording days. When performing these extended TF and CF tasks (Fig. 1b, c), Z performed ~400 correct trials/checkerboard coherence plus variable numbers of error trials in each task over the course of several daily testing sessions, resulting in ~6000 trials per task.

To test whether the difference in task performance between the two monkeys in the CF versus CFD tasks was due to the difference in their temporal structure, Z was re-trained and tested in a task with identical temporal structure to the CFD task, including a fixed 500 ms Checkerboard-observation epoch followed by a variable 400–800 ms memory-delay period. Trials were presented with seven checkerboard coherence levels (4%, 10%, 20%, 30%, 40%, 60%, and 80%; Supplementary Figure 1). These behavioral data were collected after all neural data had been collected in the TF and CF tasks.

**Recording chamber implantation and neural data collection**. Prior to surgical preparation for neural recordings, an anatomical MRI scan was made of Z's head to provide images of the sulcal patterns of its cerebral cortex and their location relative to small fiducial-marker gold pins (Hybex Innovations) implanted in its skull at known stereotaxic coordinates. Z then had custom-made titanium recording chambers implanted over 18 mm diameter trephine holes made in the skull under stereotaxic control (coordinates A21, L15; Fig. 1e). The initial implant was over left PMd/M1. After neural data collection was completed, that chamber was removed and the skin opening was closed for several months while it was trained with the left arm. After training with the left arm, a second chamber was implanted over the right PMd/M1. All surgical procedures were performed using standard aseptic surgical procedures[74,75].

Neural activity was recorded using single in-house-made Corning glass-insulated platinum–iridium microelectrodes. In each daily recording session, the electrode was lowered through the dura and into the brain at a chosen electrode location within the chamber, using a Chubbuck electromechanical microdrive[76]. The electrode was advanced while Z performed the 1T and 2T tasks using eight reach target directions, to search for units that showed strong and directionally tuned activity in the tasks. Once a task-related unit was isolated, the target that elicited the strongest task-related activity changes during initial screening tests with eight reach directions in the 1T and 2T tasks was designated as its PD. Data files were next collected for short trial blocks (20–40 trials) in the PD and the opposite target direction (oPD) in the 1T and the 2T tasks. Data were then collected in the TF, CF, and TFCD tasks in pseudo-random order, to collect at least one and ideally two complete data files in each task. A unit was retained for detailed analysis if its isolation and task-related activity remained stable throughout the recording session, and data were collected successfully from at least one complete data file for the TF and CF tasks.

**Neural activity pre-processing**. The spike waveforms of the recorded unit were isolated and their times were digitized in real time at 1 ms resolution using a two-window spike–amplitude discriminator. For most analyses, the digitized spike times in each trial were converted into a continuous pseudo-analog signal using the partial inter-spike intervals that fell within each sequential time bin in a trial[44,49,74,76]. Time bin durations varied from 1 to 20 ms in different analyses. Single-trial data were divided into time windows of fixed lengths (e.g. 5 or 20 ms) or into variable-duration sequential trial epochs for different analyses. An automatic algorithm counted the numbers of whole and fractional inter-spike intervals that fell within the time window or trial epoch. If an inter-spike interval spanned two or more contiguous time windows or epochs, each window or epoch received a fractional count proportional to the fraction of the inter-spike interval that fell within its boundaries. The partial-spike scores were then converted to single-trial spikes/s discharge rates by normalizing for the duration of the time window or trial epoch. Mean cell response histograms were generated by aligning all the single-trial data to different time points in the trial, summing the single-trial discharge rates in corresponding time bins across all trials and then normalizing by the numbers of trials.

All data files were also pre-processed by an automatic algorithm to identify the time of the movement onset (the Reaction Time), and any changes in direction during the reaching movement. The results of this automated analysis of reach kinematics were visually verified for every trial and were corrected manually when necessary (see ref. [49] for details).

**Quantification and statistical analyses (for both T and Z)**. Data were analyzed using custom scripts in MATLAB (The Mathworks, Inc.) developed and shared by the two labs. Note that both the box-car smoothing and inter-spike interval approaches used in the two labs to convert spike times into firing rates are well-established. All results were fundamentally identical when the two different discharge-rate conversion algorithms were applied to the same data files (results not shown).

**Psychophysical threshold**. Psychophysical performance was fit to a cumulative Weibull function using the *fit* function of the MATLAB 2018A curve-fitting toolbox. Where $x$ is checkerboard coherence, and $p$ is the proportion of correct responses:

$$p = 1 - 0.5 e^{-\left(\frac{x}{\alpha}\right)^{\beta}}$$

The $\alpha$ parameter is the psychophysical threshold, as it is equivalent to the checkerboard coherence at which performance reaches 81.6% correct responses.

**Rapid response changes evoked by the appearance of the first visual cue**. To identify abrupt overt response changes in single-unit activity elicited by the first visual cue in each task, we aligned all single-trial data for each unit to the onset of the first cue. Activity was pooled across all trials without regard to checkerboard coherences or eventual reach directions. We then tested the distributions of single-trial activity in each 20 ms bin against the 20 ms bin two bins previously (for instance, the 40–60 ms bin vs. the 0–20 ms bin), incremented in 20 ms steps from 0 to 600 ms after the appearance of the first cue (Wilcoxon signed rank test, $p < 0.01$). A unit was identified as having a significant abrupt change in activity if it showed a significant change in two consecutive time steps, i.e., significant activity differences spanning an 80 ms time window, and we noted the time bin in which the first significant rapid response change occurred.

**Slope of choice selectivity signal**. To calculate the directional choice selectivity signal[26,51], we aligned the single-trial neural activity to the onset of the first and second visual cues in each trial, using both correct and incorrect target-choice trials. We then averaged the single-trial firing rate traces for left/right or PD/oPD reaches separately, for each of the checkerboard coherences. The absolute difference in these left and right averages represents the directional choice selectivity signal for each unit (spikes/s) as it evolves over time. Choice selectivity signals were calculated during a time window from 0 to 300 ms after the first and second visual cues in each task. We then used the MATLAB fit function (Matlab 2018A curve-fitting toolbox, The Mathworks Inc.) to estimate the onset time and slope of a linear change in activity after the appearance of each visual cue.

**Repeated-measures ANOVA**. A repeated-measures three-way ANOVA (IBM SPSS version 24) was performed on the mean single-trial discharge rates recorded in each trial epoch. Main factors were chosen reach direction (Direction, D), unsigned checkerboard color coherence independent of dominant color (Strength, S), and checkerboard dominant color (Color, C). The acceptable significance level was set at $p < 0.01$ (Bonferroni corrected), and the Greenhouse–Geisser correction was used whenever the sphericity assumption was violated. The ANOVA was done using only trials in which the monkeys chose the correct target, to avoid adding a fourth factor (correct/incorrect choice) to the ANOVA design. The trial epochs included the Center-Hold epoch before the first cues appeared, the Targets-observation epoch before the checkerboard appeared (TF), the Checkerboard-observation epoch before the targets appeared (CF) or the checkerboard was extinguished (CFD), the Checkerboard-RT (TF) and Targets-RT (CF/CFD) epochs from the appearance of the second cue to the onset of movement, the Movement epoch for the duration of the movement from its onset until the arm reached the target, and the Target-hold epoch after target entry to the end of the trial (all tasks).

Unsigned evidence Strength here is a measure of the relative strength of the dominant color of the checkerboards without consideration of its actual color or its level of support for a particular target direction, as contrasted with the linear regression analysis (Fig. 5). It may also be predictive of the level of confidence that the monkeys could have that their perceptual/motor decision in response to a given checkerboard coherence will be correct, based on lengthy experience with the associated success rates (Fig. 1b;[57,60], Montanède and Kalaska, 2017, SfN Abstract).

**Linear regression analysis of the time course of correlations with different task factors**. We assessed to what degree variability in each unit's activity could be explained by checkerboard parameters and the animal's choice behavior as a function of time in each trial. We created two matrices of the single-trial firing rates, $y$, of each unit's neural activity in each task calculated in non-overlapping 20 ms time bins, aligned to either the appearance of the first visual cue or the second visual cue in each trial. We also created a design matrix of predictors, $X$, that included a bias term (all ones) and four task parameter predictors, including the direction of the chosen target independent of its color (e.g., left/oPD reach = −1, right/PD reach = +1), the color of the chosen target independent of its direction (e.g., red/blue = −1, green/yellow = +1), the signed checkerboard color coherence favoring the color of a target independent of its direction (ranging from −100%

(red/blue) to +100% (green/yellow); the amount of color evidence for one colored target over the other), and the signed checkerboard coherence strength favoring a direction of target choice independent of its color (from −100% for left/oPD to +100% for right/PD; the amount of color-independent evidence supporting one reach direction over the other). The last predictor requires knowledge of the specific target location–color conjunctions in each trial. Data from trials in which the monkeys chose the correct or incorrect target in each trial were included in this regression analysis so that the color of the chosen target can serve as a surrogate of the monkeys' perceptual interpretation of the color evidence provided by the checkerboard independent of its correct dominant color. Results were similar with and without predictor normalization (e.g., −1 to +1 for all predictors).

For each unit, the firing rate matrix was regressed against the design matrix using the Matlab function *regress* (Matlab 2018A, the Mathworks Inc.) to yield predictor weights and confidence intervals for each of those weights at each 20 ms time step, using an alpha value of a = 0.001. If the confidence interval for a predictor's weight did not include 0, then some of the variability in firing rate at this time point was significantly explained by variability in this predictor. For each predictor, we calculated the proportion of units in the population at each 20 ms time step for which the predictor's confidence interval does not include 0. The time series of significant counts for each predictor reflects how the impact of that predictor on single-unit neural activity across the population evolved in time during a trial.

### Ideal-observer analysis of the presence and time course of significant detectability of different task factors in the activity of the neural sample population.

We performed a receiver operating characteristic (ROC) analysis at successive 20 ms time intervals to assess the ability of an ideal observer to determine either the dominant color of the checkerboard or the direction of the chosen reaching movement from the distributions of recorded neural activity at different moments in time in a trial, in each task separately. For each unit, data from trials with both correct and incorrect target choices in each task were sorted into two groups according to the dominant color of the checkerboard or the chosen reach direction in each trial pooled across all checkerboard coherence levels. Single-trial firing rates were calculated at 20 ms time steps relative to the appearance of either the first or second cue for each of the two groups of trials. The two distributions of firing rates in each time bin were used to calculate the area under the ROC curve (AUC) at that time step for a given unit. This provided a time series of AUC measures for each unit for either the checkerboard color or chosen reach direction. An AUC value of 0.5 indicates an inability to differentiate the two data distributions for either the two colors or the two reach directions, while a value of 1 indicates a perfect ability to distinguish the two. This was repeated for each unit in the sample neural populations in each task (for T, only units tested in both the TF and CFD tasks were used). This yielded distributions of the AUC measures for the sample populations at each 20 ms time step for each comparison (checkerboard color or chosen reach direction). To determine whether an ideal observer of the neural activity could show an improvement in their ability to distinguish between the checkerboard colors or chosen reach directions at each time step, we compared the distributions of AUC values in each 20 ms bin against the AUC values calculated in a baseline time step −200 to −180 ms before the onset of the first visual cue in each trial (Wilcoxon 1-tailed signed-rank test, p < 0.001).

The ROC analysis was also used to test for a difference in the onset latency of an improvement in the detectability of the chosen reach direction between the two tasks. For T, the AUC values were recalculated at 1 ms time steps, starting −200 ms before the appearance of the second visual cue, and ending 600 ms after its appearance. The distributions of AUC values over units at each time step were tested against a baseline time step −200 ms before the appearance of the second cue (Wilcoxon 1-tailed signed rank test, significance threshold = 0.05/801 = 6.2E−05). The onset latency for each task was identified as the first time step after the appearance of the second visual cue that had a significant increase in AUC values and was followed by 49 ms with a significant increase (i.e., 50 ms of uninterrupted significantly larger AUC values compared to the pre-cue baseline activity). The same procedure was used for Z, but at 10 ms resolution.

To determine if the difference in latencies between tasks was significant, we used bootstrapping to create a null distribution of latency differences: we resampled units with replacement 1000 times, randomly re-assigned their ROC data task labels, and then calculated the onset latency using the re-assigned AUC distributions as described above. Note that for T, we only used units that were recorded in both tasks. This generated a null distribution of 1000 simulated latency differences that could have occurred if the task condition did not have a systematic effect on response latencies. We then counted the number of times the 1000 bootstrap-simulated latency differences was greater than the observed inter-task latency difference for the actual data sets, and calculated the p-value as described in ref. [52].

**Reporting summary**. Further information on experimental design is available in the Nature Research Reporting Summary linked to this article.

## Data availability

The data that support the findings of this study are available from the corresponding author upon reasonable request.

## Code availability

The code that analyzed the neural data of this study are available from the corresponding author upon reasonable request.

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

## Acknowledgements

Support for this project was provided by the following sources: M.W.: US DoD, Air Force Office of Scientific Research, National Defense Science and Engineering Graduate (NDSEG) Fellowship, 32 CFR 168a. C.M., J.F.K.: Canadian Institutes of Health Research operating grants MOP-97944 and MOP-142220, and an infrastructure grant from the FRSQ to the Groupe de recherche sur le système nerveux central. C.C.: NIH/NINDS K99 grant NS092972 and R00 grant 4R00NS092972-03 and Howard Hughes Medical Institute. D.P.: the Champalimaud Foundation, Portugal, and Howard Hughes Medical Institute. K.V.S.: NIH/NINDS Transformative Research Award R01NS076460, NIH/NIMH Transformative Research Award R01MH09964703, NIH Director's Pioneer Award 8DP1HD075623, DARPA Biological Technology Office (BTO) REPAIR award N66001-10-C-2010, DARPA BTO NeuroFAST award W911NF-14-2-0013, the Simons Foundation Collaboration on the Global Brain awards 325380 and 543045, and the Howard Hughes Medical Institute. We thank Dr. William T. Newsome for constructive comments on the manuscript and discussions throughout the course of this work.

## Author contributions

Study conception and design: M.W., C.M., C.C., K.V.S., and J.F.K. Data acquisition and analysis: M.W., C.M., C.C., and J.F.K. Manuscript preparation and editing: M.W., C.M., C.C., D.P., K.V.S., and J.F.K. Joint supervision of project and senior co-authors: K.V.S. and J.F.K.

## Additional information

**Competing interests:** The authors declare no competing interests.

