## [Peer Review File · Nature Communications]

Reviewers' comments:

Reviewer #1 (Remarks to the Author):

The authors report on a study in which they trained monkeys to determine the dominant color in a pixelated rectangle and choose the target that matched the dominant color with an arm movement. The task was run under two conditions. In the first condition the reach targets were shown before the pixelated rectangle. In the second condition the targets were shown after the pixelated stimulus. Therefore in the first condition they could plan their reach as they inferred the dominant color, whereas in the second they could not plan their reach direction until the targets were shown. They recorded single neuron activity in PMd while the animals carried out the task. The main finding was that the neural activity did not encode the color of the stimulus. The neural activity primarily encoded the reach direction, and also the strength of the evidence that predicted the correct direction.

This is an interesting study that directly addresses a specific question. Do premotor neurons encode decisions when the motor output necessary to indicate the decision is unknown? The study was well executed and the paper is clearly written. Although the strategies of the two monkeys differed somewhat, this did not affect the conclusion about the hypothesis of interest. I thought the execution of the experiments across institutions was interesting. I have minimal comments.

Was the moving window regression done with 20 ms bins? This is quite small. Although it gives good temporal resolution, it may miss significant neurons with low firing rates. I do see the additional analyses in Supp Fig. 3. Perhaps these could be incorporated in the main text.

Reviewer #2 (Remarks to the Author):

This paper combines the results from two different monkeys, each in a different laboratory, performing a very similar pair of tasks. For each monkey the pair of tasks is nicely designed and the resulting data bolster the authors' conclusion that neural activity in dorsal premotor cortex only appears to represent the accumulation of sensory information when such information is directly indicative of the motor action to be prepared. While the result is not entirely surprising, it is an important contribution since the two aspects of a decision-making task (sensory processing and motor planning) are often inseparable. I consider the combining of work from two different labs with slightly different task designs to make the work more valuable, supporting the main conclusions more strongly than otherwise. While some aspects of the behavior differ across the two monkeys, perhaps because of differences in training or task design across the laboratories, these do not impact the primary conclusions of the paper.

I only have one significant criticism (though on a small aspect of the analyses) as follows:

I think the bootstrap test may have been carried out incorrectly. When bootstrapping, with for example 1000 resampled data sets, if the mean of the original data is superior to all but 10 of those 1000, then the significance is 1%. In general, you do not redo a significance test on original versus bootstrapped data—it looks like this may have been done to achieve the astronomically low significance values on lines 478-479. With 1000 resampled data sets, the significance should be at most 0.001. I may be wrong though as the description in the Methods (lines 1700-1717) is a little confusing and would benefit from some expansion.

There are a few different ways that bootstrapping can be used. For example, what I would expect to see done is first, all values combined from the two datasets, to produce a single distribution, with the null hypothesis being that each dataset arises from a distinct set of uncorrelated samples from this single distribution. Now, separately draw, with replacement, 1000 pairs of datasets from this distribution. For each pair, measure the mean difference in latencies. Finally, see how the actually measured mean difference in latencies compares with these 1000 resampled values. If the difference is greater than all but 50 of those 1000, then the significance is less than 5% (in a one-tailed test).

The following comments are essentially minor and mostly suggestions for improving clarity in areas where I had some initial confusion:

There is some suggestion that the 500ms viewing period could be too short in the CFD task – perhaps as a check, the authors should compare with the probability of correct response within 500ms in the TF condition (treating trials with response times > 500ms as 50-50)?

I think it worth mentioning that each monkey only ever had the same two colors to choose from given the pairs of colors are different across monkeys and one monkey had an ignorable color (that was actually a choice color of the other monkey).

I.165 “for each unit”: units are not mentioned before this point. Nor the fact that only a single putative neuron is recorded in each trial, so that the task can be performed after measuring just this one unit’s receptive field. Some foreshadowing of this would help.

I. 253 “a modified CF task” appears to be the CFD task, so why not call it CFD here?

I.274 “reduction” that was “largest” I think still worth mentioning that this was even though the weakest stimulus produced the largest response times (the large reduction not being because responses are faster at weaker stimuli, but because in the TF task they were so much slower).

I. 304 “recorded neurons” vs “units”

I. 308-9 “in each cell’s task-related preferred ...” again, be clear that this can only be done because a single cell is recorded each trial?? Or what happens to responses from other cells when targets are placed in a particular single cell’s preferred direction – are those data ignored and not analyzed?

I.328 vs 332-3: It would be nice to see some suggestion on why almost 50% of cells see an effect in monkey Z while essentially none do in monkey T. What do you think is going on here?

I.473 “a bootstrap test in 50ms bins” – this description does not convey much about what was done. Bootstrapping is used for a single distribution, with resampling with replacement an equal number of values from that distribution, then comparison of parameters to gain a confidence interval. It is unclear here which distribution of values is being resampled with replacement and then what is compared/inferred.

I.582 “improve by 5%”: please be a little more specific. This could be a big improvement, if the ceiling goes from 10% failures (i.e. 90% success) at large coherence to only 5% failures (i.e. 95% success). Is this what is meant? Or is this the average improvement across all coherence values, which could as easily be achieved by a sharpening of the slope of the psychometric function.

l.680 "in stark contrast" – I think this makes it sound like the two experiments disagree. Perhaps ameliorate by saying "These findings in LIP ... in stark contrast to our own findings in PMd." To highlight the results are compatible but refer to distinct regions.

l. 1079-1080 "less sensitive to checkerboard coherence in the CFD/CF task" Perhaps mention that this is what would be expected if the color information had already been processed and used to form a planned binary response.

Figure 5. "single neuron" – please be clear throughout the manuscript of any distinction between "unit" or "single-unit" or "putative single neuron" or "single neuron", otherwise I suggest use only one term defined once in the Methods. Please comment on why double the number of units appear to be selective for Monkey Z versus Monkey T.

Figure 6. Please comment on why the ROC change is 5-10 times greater for Monkey Z versus Monkey T.

Supplemental Table 1 runs off the paper and the legend has numbers over the text, at least in the pdf I have to look at.

l. 1437 and Figure 1. The inclusion of the irrelevant stimuli seems like it should make the task harder for Monkey Z. Yet Monkey Z appears to do better overall. This is not commented on (I think) in the text. Any thoughts about this?

I notice in the author list that for the first author who does not receive a number of "7" or "8" or "9" it is unclear which of the 3 institutions she works in.

Point-by-point response to referees

We thank the referees for their very helpful and insightful comments. In our following point-by-point response, the referees' reviews are in black text and our responses are in blue. We agree with and follow essentially all of the suggestions made on how to improve and clarify our text.

Reviewers' comments:

Reviewer #1 (Remarks to the Author):

The authors report on a study in which they trained monkeys to determine the dominant color in a pixelated rectangle and choose the target that matched the dominant color with an arm movement. The task was run under two conditions. In the first condition the reach targets were shown before the pixelated rectangle. In the second condition the targets were shown after the pixelated stimulus. Therefore in the first condition they could plan their reach as they inferred the dominant color, whereas in the second they could not plan their reach direction until the targets were shown. They recorded single neuron activity in PMd while the animals carried out the task. The main finding was that the neural activity did not encode the color of the stimulus. The neural activity primarily encoded the reach direction, and also the strength of the evidence that predicted the correct direction.

This is an interesting study that directly addresses a specific question. Do premotor neurons encode decisions when the motor output necessary to indicate the decision is unknown? The study was well executed and the paper is clearly written. Although the strategies of the two monkeys differed somewhat, this did not affect the conclusion about the hypothesis of interest. I thought the execution of the experiments across institutions was interesting. I have minimal comments.

We would like to thank the reviewer for carefully reading, and thoughtfully summarizing, our manuscript.

Was the moving window regression done with 20 ms bins? This is quite small. Although it gives good temporal resolution, it may miss significant neurons with low firing rates. I do see the additional analyses in Supp Fig. 3. Perhaps these could be incorporated in the main text.

Thank you for raising this important point. Yes, the moving window regression analysis was performed using 20 ms bins. We acknowledge the concern being raised for neurons that spike very infrequently. However, this analysis uses smoothed firing rates for single trials, rather than

counting spikes in 20 ms time bins. We believe this helps to alleviate the problem of potentially having large numbers of bins with data values of 0.

In addition, as you have pointed out, using the entire epoch duration (as in Supp Fig. 3) yields fundamentally the same results as the moving window regression analysis. We have emphasized this point in the edited main text (lines 458-465) as suggested.

Reviewer #2 (Remarks to the Author):

This paper combines the results from two different monkeys, each in a different laboratory, performing a very similar pair of tasks. For each monkey the pair of tasks is nicely designed and the resulting data bolster the authors' conclusion that neural activity in dorsal premotor cortex only appears to represent the accumulation of sensory information when such information is directly indicative of the motor action to be prepared. While the result is not entirely surprising, it is an important contribution since the two aspects of a decision-making task (sensory processing and motor planning) are often inseparable. I consider the combining of work from two different labs with slightly different task designs to make the work more valuable, supporting the main conclusions more strongly than otherwise. While some aspects of the behavior differ across the two monkeys, perhaps because of differences in training or task design across the laboratories, these do not impact the primary conclusions of the paper.

We would like to thank the reviewer for carefully reading, and insightfully summarizing, our manuscript. We are also gratified that our two-institution collaborative approach is appreciated as a strength of the report.

I only have one significant criticism (though on a small aspect of the analyses) as follows:

I think the bootstrap test may have been carried out incorrectly. When bootstrapping, with for example 1000 resampled data sets, if the mean of the original data is superior to all but 10 of those 1000, then the significance is 1%. In general, you do not redo a significance test on original versus bootstrapped data—it looks like this may have been done to achieve the astronomically low significance values on lines 478-479. With 1000 resampled data sets, the significance should be at most 0.001. I may be wrong though as the description in the Methods (lines 1700-1717) is a little confusing and would benefit from some expansion.

There are a few different ways that bootstrapping can be used. For example, what I would expect to see done is first, all values combined from the two datasets, to produce a single distribution,

with the null hypothesis being that each dataset arises from a distinct set of uncorrelated samples from this single distribution. Now, separately draw, with replacement, 1000 pairs of datasets from this distribution. For each pair, measure the mean difference in latencies. Finally, see how the actually measured mean difference in latencies compares with these 1000 resampled values. If the difference is greater than all but 50 of those 1000, then the significance is less than 5% (in a one-tailed test).

We thank the reviewer for pointing this out and we have made improvements to clarify the analysis details (in Results, lines 508-525; in Methods, lines 1804-1813). In our data, we do not have a mean onset latency across units; rather, latency is calculated across the population by testing population distributions of AUC values at each time point against a baseline time point (using a Wilcoxon 1-tailed signed rank test). If we perform this on our data, then for a given monkey, we have 1 value for each task, and this value describes the latency of the population to distinguish chosen reach direction. These numbers may be different, but are they significantly different? In our original manuscript, to get a better estimate of this latency, we use bootstrapping across units (randomly resample with replacement from our population of units) and perform the same analysis (comparing time points to the baseline time point to determine latency). We did this 1000 times to get a distribution of latencies for a given task. We then calculated a mean and SEM for our estimates of latency. None of the values in this distribution (TF latency - CFD/CF latency) were negative, indicating that for all bootstrap samples, the CFD/CF task latency was shorter than in TF latency.

You very kindly and correctly pointed out that our next step - asking if the distributions are significantly different - was in error. We have now adjusted the analysis and updated the accompanying text: for each population sample, we randomly shuffle the task labels and calculate latency for each task. These differences in latencies for each population sample form a null distribution of latency differences, which would occur if the task identity did not matter. For each monkey, our measured latency difference value was greater than all values in this null distribution, indicating that our latency difference is significant (monkey T: $p = 0.001$, monkey Z: $p = 0.001$).

In summary, in our original analysis, we were not using the bootstrap method as statistical test; we were using bootstrapping to generate simulated datasets. However, after your suggestion, we have improved upon and clarified the analysis by using the task label shuffling, and indeed using the bootstrap method as a statistical test. These edits are reflected in the main text (lines 508-525) as well as in the Methods (lines 1804-1813).

The following comments are essentially minor and mostly suggestions for improving clarity in areas where I had some initial confusion:

There is some suggestion that the 500ms viewing period could be too short in the CFD task – perhaps as a check, the authors should compare with the probability of correct response within 500ms in the TF condition (treating trials with response times > 500ms as 50-50)?

Thank you for this suggestion. We have plotted below the proportion of correct responses as a function of coherence levels for monkey T, for CFD data (checkerboard presented for 500 ms) as well as for TF data for which only trials with reaction times less than 500, 400, or 350 ms were kept (see legend).

As a reminder, in the TF task, the checkerboard disappears when the hand moves from center hold; thus, trials with reaction times less than 500 ms have the same or shorter checkerboard viewing time as in CFD. Given the high performance on TF trials shown here (even for those trials whose reaction times were less than 350 ms!), we conclude that 500 ms checkerboard viewing time is sufficient for the animal to achieve very reasonable discrimination performance. We have commented on this in the main text (lines 627-630) as well as added this as Supplemental Figure 4A (lines 1268-1275).

I think it worth mentioning that each monkey only ever had the same two colors to choose from given the pairs of colors are different across monkeys and one monkey had an ignorable color (that was actually a choice color of the other monkey).

Thank you for this suggestion. We have added a sentence clarifying this in Methods (line 1356-1360).

l.165 “for each unit”: units are not mentioned before this point. Nor the fact that only a single putative neuron is recorded in each trial, so that the task can be performed after measuring just this one unit’s receptive field. Some foreshadowing of this would help.

Thank you for bringing this to our attention. We moved that sentence to the Results section “Recordings in PMd” (lines 301-327) and placed a more general description for the task design section (lines 167-168).

l. 253 “a modified CF task” appears to be the CFD task, so why not call it CFD here?

Thank you. We have edited the text to call the task ‘CFD’ instead of ‘modified’ (line 256).

l.274 “reduction” that was “largest” I think still worth mentioning that this was even though the weakest stimulus produced the largest response times (the large reduction not being because responses are faster at weaker stimuli, but because in the TF task they were so much slower).

Thank you for this suggestion. We have added this comment (lines 276-277).

l. 304 “recorded neurons” vs “units”

We have replaced “neuron” with “unit” in the manuscript, wherever we are referring to our own data, and have added a sentence in the Results describing how most of our units are putative single neurons (line 304-309, see also Methods lines 1473-1485 for monkey T, lines 1616-1630 for monkey Z). For single electrode recordings, we were able to isolate waveforms very well; due to the nature of linear-array recordings, we could not always isolate units on every channel, and thus a proportion of units are multi-units.

l. 308-9 “in each cell’s task-related preferred ...” again, be clear that this can only be done because a single cell is recorded each trial?? Or what happens to responses from other cells when targets are placed in a particular single cell’s preferred direction – are those data ignored and not analyzed?

Thank you, we should have been clearer. This is a distinction between monkey T and Z that we have now clarified in the text. For monkey T, target locations are fixed, and we do not place targets in a cell’s preferred and nonpreferred directions. This is partly because we are performing linear array recording sessions where many units are being recorded from at once, and these units do not all share the same preferred direction.

On the other hand, recordings in monkey Z were performed with one unit per recording session, which means the cell's preferred direction can be identified, and the targets can be placed in preferred and nonpreferred directions. This difference will also bear on some of your comments later in the review about different amounts of directional selectivity between T and Z -- for T, we did not place targets in a manner so as to elicit maximum directional selectivity, whereas we did exactly that for Z. Thus, the selectivity over the population is expected to be lower for T. We have clarified this in the main text (lines 304-327).

1.328 vs 332-3: It would be nice to see some suggestion on why almost 50% of cells see an effect in monkey Z while essentially none do in monkey T. What do you think is going on here?

This is a great question that we address in our first Supplemental Material section (lines 1876-1959). In that section, we discuss several possibilities - slightly different brain regions (possible), training history differences (maybe), target location placement (most likely), cognitive strategy (maybe). Given the differences in specific task design, animal disposition, and training history in each lab, it is difficult to fully commit to one of these possibilities as the definitive reason for transient responses in monkey Z that are not present in monkey T. However, the target location placement seems one of the more likelier explanations, given the previous reports of transient potential actions when presenting targets in a unit's preferred direction. We have edited the Results text to include this possibility and direct the reader to the Supplemental Material for more information (line 351-355).

1.473 "a bootstrap test in 50ms bins" – this description does not convey much about what was done. Bootstrapping is used for a single distribution, with resampling with replacement an equal number of values from that distribution, then comparison of parameters to gain a confidence interval. It is unclear here which distribution of values is being resampled with replacement and then what is compared/inferred.

Thank you, we agree. We have corrected and clarified the analysis in the main text (lines 508-525) and in the Methods (lines 1804-1813). We were originally not using the bootstrap method as a statistical test, but were using bootstrapping to generate simulated datasets to form latency difference estimates. We have updated the analysis to create a null distribution of latency differences: we resampled units, randomly assigned task labels to ROC data from each unit, calculated latency differences for each resampled population, and then compared our actually measured difference in latency to that distribution.

1.582 "improve by 5%": please be a little more specific. This could be a big improvement, if the ceiling goes from 10% failures (i.e. 90% success) at large coherence to only 5% failures (i.e. 95% success). Is this what is meant? Or is this the average improvement across all coherence

values, which could as easily be achieved by a sharpening of the slope of the psychometric function.

Thank you for raising this important point. Correct, the approximate 5% improvement mentioned in the text is the total increased percent of correct trials across all coherence values. The specific variations in behavior over blank delay durations and checkerboard durations is not a focus of the paper; the Discussion text has been edited to reflect this (line 631-632). In the left and center plots below are data from monkey T on 9 sessions on which both stimulus and blank delay durations varied. We calculate the percent correct in 100 ms time bins of either blank delay duration or stimulus durations, stepping in 10 ms time steps over the range of durations. The percent correct, across all coherence values, is higher for shorter blank delay durations and longer stimulus durations.

In the right plot, we separate all of monkey T’s CFD trials by shorter (400-600 ms) and longer (600-800 ms) blank delay durations and plot the psychometric curve to show that the average improvement across all coherence values as delays shorten (from the left plot above) is achieved both by a sharpening of the slope as well as a decreased lapse rate. We agree that this would be of great interest to the reader, so we briefly describe this in the Discussion, where we raise these points (lines 627-634) and have included these figures in Supplemental Figure 4B,C,D (lines 1276-1286).

1.680 “in stark contrast” – I think this makes it sound like the two experiments disagree. Perhaps ameliorate by saying “These findings in LIP ... in stark contrast to our own findings in PMd.” To highlight the results are compatible but refer to distinct regions.

Thank you for pointing this out, we have adjusted the language to reflect that the findings in different regions differ, rather than disputing others’ results (line 736-737).

l. 1079-1080 “less sensitive to checkerboard coherence in the CFD/CF task” Perhaps mention that this is what would be expected if the color information had already been processed and used to form a planned binary response.

We have edited this to reinforce our interpretation of the results (line 1142-1145).

Figure 5. “single neuron” – please be clear throughout the manuscript of any distinction between “unit” or “single-unit” or “putative single neuron” or “single neuron”, otherwise I suggest use only one term defined once in the Methods. Please comment on why double the number of units appear to be selective for Monkey Z versus Monkey T.

As mentioned in response to your previous comment about line 304:

We have replaced “neuron” with “unit” in the manuscript, wherever we are referring to our own data, and have added a sentence in the Results describing how most of our units are putative single neurons (line 304-309, see also Methods lines 1473-1485 for monkey T, lines 1616-1630 for monkey Z). For single electrode recordings, we were able to isolate waveforms very well; due to the nature of linear-array recordings, we could not always isolate units on every channel, and thus a proportion of units are multi-units.

We have edited the Figure 5 legend (line 1168-1169) acknowledging that monkey Z’s units were more selective because targets were placed in preferred and non-preferred directions for a given unit. In contrast, monkey T’s targets were placed in the same location for all units, independent of any tuning information of the units.

Figure 6. Please comment on why the ROC change is 5-10 times greater for Monkey Z versus Monkey T.

Thank you for raising this point. For monkey Z, targets were placed in individual units’ preferred and non-preferred directions maximizing the direction selectivity. For monkey T, targets locations were fixed, irrespective of units’ preferred directions - this results in less differential activity for many units and thus the AUC values for the ROC analysis are smaller. We have added a description of this in the Results text (lines 445-446, 495-500) and commented on it in the Figure 6 legend (lines 1188-1190).

Supplemental Table 1 runs off the paper and the legend has numbers over the text, at least in the pdf I have to look at.

Our apologies for this happening for some reason. We have adjusted the Supplemental Table 1 so that the table does not run off the edge of the paper (lines 1288-1291).

l. 1437 and Figure 1. The inclusion of the irrelevant stimuli seems like it should make the task harder for Monkey Z. Yet Monkey Z appears to do better overall. This is not commented on (I think) in the text. Any thoughts about this?

Thank you for raising this. We comment on this very briefly in the second section of our Supplemental Materials (lines 1998-2008). First, the two animals have very comparable high performance in the TF task. In the TF task, both monkeys process checkerboard information very well, and extract salient information, despite the presence (monkey Z) or absence (monkey T) of irrelevant color information in the checkerboard. Second, we believe the reviewer might be comparing monkey Z's CF performance to monkey T's CFD performance - in which case, monkey Z appears to do better overall. However, a better comparison is between monkey T's CFD performance (red line, Figure 1B) to monkey Z's CFD performance (green line, Supp Fig 1A); these are of comparable performance, and are both worse than in TF. So, overall, monkey Z is not necessarily doing better, and it is likely that the irrelevant stimuli do not make the task harder for monkey Z because monkey Z is over trained on the task. (Side note that may be of interest: human participants have actually reported the TF task to be more challenging than the CF task. In addition, humans immediately understand that the red background is irrelevant and not a distractor; they simply ignore the red background. We would have to assume that monkey Z, with such over training, also ignores the red background and focuses on the blue and yellow sensory evidence.) We have also added a comment to this in the main text Results (lines 208-210).

I notice in the author list that for the first author who does not receive a number of "7" or "8" or "9" it is unclear which of the 3 institutions she works in.

The author list was written to indicate that affiliations 1-7 are all located at Stanford University, Stanford, CA 94305, USA (line 19). Please let us know if there are other suggestions for improving clarity here.

Reviewers' Comments:

Reviewer #2:

Remarks to the Author:

All my prior comments have been addressed well by the authors. I think this is a fine manuscript and a good, solid contribution to the field,